# Development of Effective Medical Countermeasures Against the Main Biowarfare Agents: The Importance of Antibodies

**DOI:** 10.3390/microorganisms12122622

**Published:** 2024-12-18

**Authors:** Arnaud Avril, Sophie Guillier, Christine Rasetti-Escargueil

**Affiliations:** 1Unité Interaction Hôte-Pathogène, Département Microbiologie et Maladies Infectieuses, Institut de Recherche Biomédicale des Armées, 91220 Brétigny-sur-Orge, France; 2Unité Bactériologie, Département Microbiologie et Maladies Infectieuses, Institut de Recherche Biomédicale des Armées, 91220 Brétigny-sur-Orge, France; sophie.guillier@intradef.gouv.fr; 3UMR_MD1, Inserm U1261, 91220 Brétigny sur Orge, France; 4Targeted Therapy Team, Institute for Cancer Research, 237 Fulham Road, London SW3 6JB, UK; christine.escargueil@laposte.net

**Keywords:** medical countermeasure, biodefense, botulinum neurotoxins (BoNTs), antitoxin, antibodies, therapeutic antibody, vaccines, toxoids, ricin, anthrax

## Abstract

The COVID-19 and mpox crisis has reminded the world of the potentially catastrophic consequences of biological agents. Aside from the natural risk, biological agents can also be weaponized or used for bioterrorism. Dissemination in a population or among livestock could be used to destabilize a nation by creating a climate of terror, by negatively impacting the economy and undermining institutions. The Centers for Disease Control and Prevention (CDC) classify biological agents into three categories (A or Tier 1, B and C) according to the risk they pose to the public and national security. Category A or Tier 1 consists of the six pathogens with the highest risk to the population (*Bacillus anthracis*, *Yersinia pestis*, *Francisella tularensis*, botulinum neurotoxins, smallpox and viral hemorrhagic fevers). Several medical countermeasures, such as vaccines, antibodies and chemical drugs, have been developed to prevent or cure the diseases induced by these pathogens. This review presents an overview of the primary medical countermeasures, and in particular, of the antibodies available against the six pathogens on the CDC’s Tier 1 agents list, as well as against ricin.

## 1. Introduction

### 1.1. Definitions

A biowarfare agent could be defined as a biological pathogen or a toxin that could be used for the development of a bioweapon. Biowarfare agents can be used to kill, harm or incapacitate humans, animals or plants. In the context of terrorism, the term biothreat agent could also be used. A bioweapon could be defined as an intentionally harmful combination of one or more biological agents with a vector (bomb, rocket, nebulizer, animal, etc.) A bioweapon could be used directly to incapacitate or kill a single person or population, or indirectly to destabilize a nation by targeting livestock and agriculture.

### 1.2. History of Biological Agents as Bioweapons

The first utilization of a bioweapon may go back to 1350 BCE when the Hittites left animals infected with tularemia, in villages which they had plundered, to kill off the local population (Table 1) [1]. When Mongols were besieging the city of Kaffa in 1346 CE, they catapulted the corpses of plague victims over the bulwark [2]. The civilians were physically and psychologically impacted. A side effect of this attack was that both healthy and sick people ran away from the city, inducing or worsening a five-year plague epidemic that killed about a third of the European population. This historical event illustrates the potential close connection between intentional and natural biological risk. It also highlights the complexity of controlling a biological attack.

During World War I, numerous nations initiated national programs aimed at developing biological weapons for both defensive and offensive purposes, although the distinction between the two can often be complicated. An example of such a program is the Soviet initiative known as Biopreparat [2]. Established in 1919 and concluded in 1991, this program operated with an annual budget estimated at US$1 billion dollars. It militarized eight biological agents—including *variola virus*, *Yersinia pestis*, *Francisella tularensis*, glanders-like pathogens, equine encephalitis, *Bacillus anthracis*, Q-fever and Marburg virus—and produced them in large quantities. Several of these agents were genetically modified to enhance their pathogenicity or resistance to available treatments. A significant accident occurred in 1979 at an anthrax spore production facility. Following maintenance work, the backup particle filters in the air conditioning system were not correctly replaced, leading to the release of anthrax spores into the environment, which caused a severe outbreak (66 people died from inhalational anthrax) [3]. This incident highlights the potentially catastrophic outcomes of handling biological agents, whether for military or conventional research purposes. Another notable state-sponsored program was Japan’s Unit 731, initiated in 1925 with around 3000 personnel. This unit produced substantial quantities of pathogens and subjected thousands of Chinese prisoners to these agents without treatment to study the resulting diseases [4]. Some of these biological agents were also spread in China, both directly by contaminating water sources and indirectly using vectors like fleas or bombs, resulting in an estimated 580,000 deaths [5]. In 1942, the use of bioweapons against Chinese forces also led to the deaths of over 1700 Japanese soldiers, demonstrating the unpredictable nature of biological agents [6].

A significant bioterrorism event occurred in September 2001 when anthrax spores were mailed to American government officials and to media, resulting in eleven infections and five deaths despite prompt treatment (antibiotics and intensive care). In 2008, FBI investigations identified Bruce Ivins, a scientist at the United States Army Medical Research Institute of Infectious Diseases (USAMRIID), as the prime suspect. Ivins allegedly stole *B. anthracis* spores from the lab, and, being vaccinated, he could handle the material safely. The FBI investigation revealed several biosecurity weaknesses, including relatively easy access to dangerous pathogens and insufficient monitoring of the mental health of personnel working with such agents. This case illustrates how a single individual, if skilled and determined, can develop and deploy a bioweapon successfully [7].

**Table 1 microorganisms-12-02622-t001:** Examples of suspected or recognized use of a pathogen as a biowarfare agent.

Year	Event	Reference
1350 BCE	Hittites leave animals contaminated with tularemia in plundered villages.	[8]
4th Century BCE	According to Herodotus, Scythian archers infect their arrows by dipping them into decomposing corpses.	[8]
1155	Emperor Barbarossa poisons water wells with human bodies.	[8]
1346	Mongols catapult bodies of plague victims over the fortifications of Kaffa (Feodosia, Crimean Peninsula).	[8]
1422	Prince Zygmunt Korybutovic hurls corpses of plague-stricken soldiers, dead cows and excrement during the siege of Karistejn (the modern-day Czech Republic).	[8]
1495	The Spanish mix wine with the blood of leprosy patients to sell to their French foe (Naples, Italy).	[8]
1650	Polish General Kazimierz Siemienowicz fires hollow artillery shells filled with the saliva of rabid dogs.	[8]
1763	British troops give smallpox-infected blankets to the Native Americans.	[8]
1797	Napoleon floods the plains around Mantua, Italy, to enhance the spread of malaria.	[8]
1785	Tunisians throw plague-infected clothing into the Christian-held city of La Calle (modern-day Algeria).	[8]
1863	Confederates sell clothing from yellow fever and smallpox patients to Union troops, USA.	[8]
1942	Japanese Unit 731 uses bioweapons against the Chinese.	[8]
27 May 1942	Jan Kubis, a Czech member of the Resistance uses a grenade coated with botulinum toxins to kill Nazi General Reinhard Heydrich.	[8]
1978	The KGB kills Georgi Markov in London with a system hidden inside an umbrella that injects spheres containing ricin.	[8]
1984	The Rajneesh cult contaminates salad bars with Salmonella typhimurium in Dallas (USA).	[8]
April 1990	The Aum Shinrikyō sect tries to spray what they think is botulinum toxin throughout Tokyo, Yokohama, Yokosuka and Narita using nebulizers placed in trucks.	[8]
June 1993	The Aum Shinrikyō sect tries to spread anthrax spores from trucks.	[8]
Summer 1993	The Aum Shinrikyō sect tries to spread anthrax spores in Tokyo from a rooftop.	[8]
Autumn 2001	Letters containing anthrax spores are sent to American officials (probably sent by the researcher Bruce Ivins).	[8]
April 2013	Three letters containing ricin are sent to the President of the USA and to American officials.	[8]
2016	DAESH plans to use anthrax in a mall in Nairobi.	[8]
2018	Terrorists arrested in Cologne (Germany) with explosive devices that contain 84.3 mg of ricin.	[9]
2018 and 2020	Letters containing ricin are sent to President Donald Trump.	[10,11]

### 1.3. International Regulation of Research on Biological Agents

It is sometimes complicated to make a balance between the good and the dark side of some research projects (dual research) [12]. The first regulation of bacteriological weapons development dates back to the 1925 Geneva Conference for the Supervision of the International Traffic in Arms. It banned the use of bacteriological weapons, but it did not ban their development, and several countries did not ratify the Convention.

In 1972, the Biological Weapons Convention was signed by several countries under the supervision of the United Nations Office for Disarmament Affairs. This Convention prohibits the development, production, acquisition, transfer, stockpiling and use of biological and toxin weapons. It was the first multilateral disarmament treaty banning an entire category of weapons of mass destruction. This treaty is more restrictive than the Geneva Protocol, but it still does not include sanctions in case of infractions. Additionally, of the 183 state parties, 4 have still not signed the treaty. Additionally, 10 states have still not joined the convention [13].

In 1975, the first conference took place in Asilomar (Pacific Grove, CA, USA) with 150 scientists, to discuss the pros and the cons of dual-use research. Unfortunately, this Conference was purely scientific, no regulations were recommended, and it did not take into account all aspects of such research (such as environmental aspects). Only recommendations concerning the confinement of dangerous biological agents, such as GMOs (as they were historically considered as very dangerous), were given. This Conference is now outdated, due to recent progress in molecular biology and artificial intelligence.

On 28 April 2004, the United Nations Security Council unanimously adopted resolution 1540 (2004) which establishes that the proliferation of nuclear, chemical and biological weapons and their means of delivery constitutes a threat to international peace and security [14]. The resolution obliges states to refrain from supporting by any means non-state actors from developing, acquiring, manufacturing, possessing, transporting, transferring or using nuclear, chemical or biological weapons and their means of delivery. This resolution imposes binding obligations on all states to adopt suitable legislation and to establish appropriate domestic controls to prevent the trafficking of illicit materials. On 22 April 2021, the Security Council unanimously adopted resolution 2572 (2021) to extend the mandate of the 1540 Committee until 28 February 2022.

Aside from internal regulation, research centers and scientific journals also have an ethical role. Indeed, scientific progress is closely tied to the sharing of results with the scientific community. In the last two decades, progress in the genetic and molecular engineering of biological agents and in artificial intelligence has opened up a world of opportunity for scientists. Such progress makes the comprehension of biological agents and the development of medical countermeasures easier and faster. A dark side of this technical progress is that the same technology may be used for the development of bioweapons; this research is referred to as dual-use research of concern. These technologies also raise ethical concerns about the intentional modification of pathogens, even if it is done for good reasons. As an example, gain of function could be useful for the development of new therapeutics, but the intentional or natural release of such “improved” pathogens could be dramatic.

### 1.4. Lessons Learned from the COVID-19 and Mpox Crises

The COVID-19 crisis revealed that the world is still vulnerable to biological threats. At present, the origin of the severe acute respiratory syndrome-coronavirus 2 (SARS-CoV-2) virus, which caused the COVID-19 pandemic, has yet to be fully determined. Several hypotheses are under investigation to identify the origin of the virus [15,16,17,18,19,20,21,22,23]. Whether the origin was natural, accidental or intentional, this crisis underlined the weakness of the world population when exposed to some biological agents. Currently, there is evidence for both natural and human (accidental or intentional) origin. There is scientific evidence for a natural origin: as an example, SARS-CoV-2 could have derived from another coronavirus, because such evolution is natural and common. Nevertheless, according to another hypothesis, the mutations observed in the SARS-CoV-2 sequences could be the result of the genetic engineering of other coronaviruses, even if mutation can also occur naturally. Although there is evidence to support zoonotic transfer, inconclusive reports keep the laboratory leak hypothesis alive. All hypotheses must be considered possible until there is proof to the contrary; investigators should consider all hypotheses with the same rigor. It is imperative to reach a factual conclusion to prevent future pandemics.

Considering the major consequences of the COVID-19 crisis, national preparedness is essential. For example, NATO wrote a report on bioterrorism evaluating the probability that the knowledge required to produce bioweapons was available for bioterrorists and malicious countries. It also presented the tools implemented by NATO and state parties for preparedness [24]. Nevertheless, it has been estimated that only a quarter of the world’s 59 BSL-4 labs are located in countries that score well in a ranking of international biosafety and biosecurity metrics drawn up by the US-based Nuclear Threat Initiative, although this study has been the subject of some debate [25]. The index measures whether countries have legislation, regulation, oversight agencies and other facets of sound biorisk management. Furthermore, only three of the countries with BSL-4 labs have policies on dual-use research (Canada, the United States of America and the United Kingdom) [26], meaning that the vast majority of countries with such laboratories do not conduct oversight of the type of high-risk, gain-of-function research that makes pathogens more dangerous and that has been a central feature in the COVID-19 origin debate.

After the COVID-19 crisis, the mpox outbreak demonstrated again that a virus can rapidly diffuse worldwide. In January 2024, 92,546 cases were laboratory confirmed in 117 countries, and 170 deaths had been recorded. The real number of infections is probably higher, due to the difficulties encountered in some countries in making laboratory diagnoses. This was the first time that so many infections had occurred outside of Africa. Unlike SARS-CoV-2, vaccines and drugs were available for smallpox and stockpiled. The challenge during this crisis was to confirm that the vaccines (IMVANEX^®^, JYNNEOSTM and ACAM2000^®^) and Tecovirimat were also effective against the monkeypox virus [25,27]. Mpox was initially a zoonotic disease, but this outbreak is characterized by a modification of its epidemiology (sexual transmission) [28]. A thorough study of this outbreak is essential for preventing future pandemics involving other pathogens. Particularly, it is essential to be prepared for the emergence of an unknown pathogen (generally referred as “pathogen X”). This preparedness particularly involved the development of pipelines and platforms that are flexible.

### 1.5. Antibodies as Ideal Medical Countermeasures

Diseases induced by biowarfare agents can require long-term treatment, sometimes in an intensive care unit. Such medical care is very expensive, and capacity is very limited. In the context of a bioterrorist attack, these medical capacities can rapidly become overloaded. The development of medical countermeasures is essential to avoid the use of intensive care units, or to decrease hospitalization time. Generally, biowarfare agents cause neglected/rare diseases, and few medical countermeasures are available. In some cases, such as ricin intoxication, there are no approved drugs. Depending on the disease, antibodies, antibiotics or antivirals can be administered alone or in synergistic combination. The first therapeutic antibody was approved almost four decades ago (1986) [29]. Antibodies are the molecules of choice for the diagnosis and treatment of these diseases. They are highly specific, quick to develop, relatively easy to produce and they are generally much more well tolerated (with few off-target effects) than antiviral molecules. Additionally, they have a better success rate in clinical trials [30,31,32]. Crescioli et al. have estimated that antibodies have approval rates of between 14% and 32%, with higher rates associated with antibodies developed for non-cancer indications [30]. Antibodies are particularly effective in neutralizing toxins such as ricin, botulinum neurotoxins and anthrax toxin. High affinity is required to effectively neutralize toxins with toxicity levels in the ng·kg^−1^ range. In addition, when contamination occurs via the toxin alone, antibiotics are ineffective. Antibodies are also of particular interest, even when antibiotics are available, because they can act in synergy. This represents an alternative treatment against antibiotic-resistant strains, as it has been observed with Tecovirimat. Several generations of monoclonal then recombinant antibodies were developed (chimeric, humanized and fully human antibodies) to optimize their clinical tolerance. Currently, 4 murine, 27 chimeric, 59 humanized and 72 fully human antibodies have been approved for therapeutic use. Some of them have been approved for the treatment of several diseases. The global therapeutic monoclonal antibody market was valued at approximately US$247 billion in 2023 and is expected to rise to US$479 billion by 2028. Even though recombinant antibodies are now the gold standard, polyclonal antibodies are still important because they offer broad neutralization of bacteria and virus particles, and they can be developed more quickly. The major limitations for polyclonal antibodies are that they are generally of animal origin and, as such, can be less well tolerated, and that the proportion of neutralizing antibodies in each batch is variable. When they are of human origin, extensive controls are required to prevent the transmission of pathogens that can be disseminated by the blood. Oligoclonal antibodies represent a good compromise between monoclonal/recombinant antibodies and polyclonal antibodies. It is indeed possible and important for clinical trials and industrialization, to produce a mixture of well-characterized antibodies directed against a specific target.

### 1.6. Objective of This Review

The aim of this review is to provide an overview of the therapeutics available against the principal biowarfare agents. Each pathogen could be the subject of a particular review. The objective of this review is to provide an essential overview of the state of the art concerning the research on these neglected pathogens. The main objective is to focus on antibodies because they are molecules that are particularly suitable in the context of biodefense. Data concerning other molecules (antivirals, antibiotics, etc.,) were also provided to underline their weakness or, on the contrary, to show their potential synergy with antibodies. This review will demonstrate that the development of antibodies is essential for public health and biodefense.

## 2. Anthrax

### 2.1. Background

Anthrax is caused by *Bacillus anthracis*, a Gram-positive, spore-forming bacteria. Anthrax is a lethal disease that infects animals such as cows, but that can also naturally infect humans [33]. Four forms of anthrax have been described in humans: cutaneous, intestinal, inhalation and injection anthrax. All of them are lethal for humans in the absence of treatment. Cutaneous anthrax is at the origin of almost all natural infections (95%) but is almost not lethal (1% lethality if treated). The lethality of pulmonary anthrax is about 45–80% when treated, and almost always lethal when untreated [34]. Gastrointestinal anthrax develops after consuming contaminated meat. Finally, contamination may also occur via the injection of a drug contaminated by the spores [35]. The multiplication of the bacteria occurs in the organs. *B. anthracis* can cross the blood–brain barrier. In more than 50% of human cases and in experimental non-human primate (NHP) models, Central Nervous System (CNS) infection is typically associated with meningeal hemorrhage (“cardinal’s cap”) [36]. When left untreated, the human mean lethal dose (LD_50_) is estimated to be between 8000 and 10,000 inhaled spores [37]. *B. anthracis* has two virulence factors, encoded by the pXO1 and pXO2 plasmids [38]. *pXO1* encodes for a poly-γ-D-glutamic acid capsule that protects it from phagocytosis. *pXO2* encodes for two anthrax exotoxins. These toxins are targets of choice for antibodies and anti-drugs. The lethal toxin (LT) is composed of the protective antigen (PA) and the lethal factor (LF). The edema toxin (ET) is composed of the PA and the edema factor (EF). PA83 binds to ubiquitous host cell membrane receptors (ANTXR1–TEM8 or ANTRX2–CMG2) and is then cleaved by a cell-associated furin-like protease to form PA63, which then oligomerizes with other PA63 molecules to form a heptamer. The heptamer forms a prepore structure to which LF or EF bind to form a lethal toxin or an edema toxin, respectively. Once formed, the whole complex is internalized by endocytosis, and LF and EF are translocated into the cytosol. LF is a protease that catalyzes the hydrolysis of the MAPKK, resulting in the cell apoptosis. EF is a calmodulin-dependent adenylate cyclase that greatly increases the levels of cAMP in the cell. The increase in cAMP induces an edema (perturbation of the water homeostasis), causes a perturbation in the intracellular signaling pathways and impairs macrophage function (immune system’s evasion).

In Europe, between 2009 and 2011, an anthrax outbreak occurred, characterized by soft-tissue infections. More than 50 confirmed cases were reported and all of them were in people with drug abuse issues. Investigations identified the origin of the contamination: Anthrax spores were found in heroin. This outbreak emphasized the consequences of a massive “natural” contamination [39]. As presented previously, in September 2001, five people died during a bioterrorist attack in the USA, highlighting anthrax’s potential to be used as a bioweapon.

### 2.2. Therapy

#### 2.2.1. Antibodies

According to the CDC, antibiotics and antibodies have to be administrated simultaneously, despite recent studies that have questioned the efficacy of antibodies [40,41,42,43]. Despite the rapid administration of antibiotics, a toxemia can occur, leading to death.

Three anthrax antibodies have been approved for use by the US Food and Drug Administration (FDA). Obiltoxaximab (Anthim) and Raxibacumab are monoclonal antibodies. Anthrasil (intravenous anthrax immune globulin AIG-IV) is composed of polyclonal IgG isolated from vaccinated humans. These antibodies are stockpiled in the USA in case of a bioterrorist attack. Because it is not feasible or ethical to realize clinical trials in humans with inhalational anthrax, the efficacy of antibodies for treatment and prophylaxis has been studied in multiple animal models, such as the cynomolgus macaques and New Zealand white rabbit (WNZ) models of inhalational anthrax (FDA Animal Rule). Of the three anthrax antitoxin agents now approved for use in the US, only AIG has actually been administered to a group of infected patients.

Raxibacumab (Abthrax) was the first recombinant antibody to obtain FDA approval (FDA 2012 and EMA 2014) for the prevention and treatment of inhalational anthrax. It is a fully human IgG1 that binds the PA with 2.78 nM affinity. Raxibacumab inhibits the pore formation. In therapy, Raxibacumab has to be administered intravenously at 80 mg·kg^−1^ in children below 15 kg and at only 40 mg·kg^−1^ over 15 kg, after Diphenhydramine premedication [44].

The effectiveness of Raxibacumab was assessed in the New Zealand white rabbit (WNZ) and cynomolgus macaque models, and its safety was assessed in healthy volunteers [45]. Rabbits and macaques were challenged with 200 LD_50_ of aerosolized anthrax Ames spores and treated with a single bolus (IV) of 40 mg·kg^−1^ of Raxibacumab after the detection of PA subunits in the serum or following a 1.1 °C rise in temperature. With this protocol, 44.4% of the rabbits and 64.3% of the macaques survived (vs 0% in the placebo group). When rabbits (n = 12/group) were administered Raxibacumab prophylactically and subcutaneously at 5, 10 or 20 mg·kg^−1^ 2 days prior to exposure, or concurrently and intravenously at 40 mg·kg^−1^ and challenged with 100 LD_50_ aerosolized Ames spores, survival rates of 40%, 83%, 83% and 100% were noted respectively, compared to 0% with placebo. Three clinical trials evaluated the safety of Raxibacumab. A total of 326 healthy volunteers were treated with one or two doses of 40 mg·kg^−1^ of Raxibacumab or placebo, alone or in combination with Ciprofloxacin. The trial was stopped for only four subjects (1.2%) that suffered serious adverse reactions.

In 1988, Obiltoxaximab (Anthim) was isolated and approved by the FDA and EMA in 2016 and 2018, respectively. Obiltoxaximab is an antibody composed of mice variable domains chimerized on human constant domains. It was optimized to increase its affinity, and the T-cell epitopes were identified and removed through the combined use of molecular and immunological approaches. The efficacy of the treatment and the prophylaxis of inhalational anthrax was demonstrated in studies conducted on animals and based on the survival rates at the end of the studies [46,47]. These studies tested the efficacy of ANTHIM compared to placebo and the efficacy of ANTHIM in combination with antibacterial drugs relative to the antibacterial drugs alone. Two studies in WNZ rabbits and two studies in cynomolgus macaques evaluated treatment with a single dose of IV ANTHIM 16 mg·kg^−1^ compared to placebo in animals with systemic anthrax. Treatment with ANTHIM alone resulted in a statistically significant improvement in survival relative to placebo in both species. The survival rate was up to 93% with ANTHIM compared to 0% with placebo in the rabbits and up to 47% with ANTHIM compared to up to 6% with placebo in macaques [48]. The safety of Obiltoxaximab was evaluated in 320 healthy human volunteers and serious adverse effects were rare.

Anthrasil (Anthrax Immune Globulin Intravenous, AIGIV) is purified from the plasma of individuals vaccinated against anthrax with anthrax vaccine adsorbed (AVA) [49,50]. The safety of the product was tested in 74 healthy human volunteers and serious side effects were rare. In the clinical study, safety, tolerability and PK were evaluated with 30 volunteers per cohort. The elimination half-life of Anthrasil in humans following intravenous infusion was estimated to be approximately 24 days, and the dose proportionality was observed. The efficacy of Anthrasil was assessed in animals exposed via inhalation to aerosolized *B. anthracis* spores. In a time-based treatment study, NZR were challenged with an aerosol containing 204 ± 47 LD_50_ (=2.2 × 10^7^ ± 5.4 × 10^6^ CFU/animal). A 21.3 mg·kg^−1^ dose of Anthrasil induced the survival of 100% or 39% of rabbits when administered 12 h or 24 h post-infection, respectively. No significant protection was observed after 24 h, underscoring the need for early intervention. The efficacy of Anthrasil was also assessed in 60 cynomolgus macaques exposed to ~200 LD_50_ aerosolized anthrax spores. Treatment with placebo or with Anthrasil was initiated after animals became toxemic (positive for PA detection in serum samples). After 88 days, survival was 0% in the placebo group versus 36%, 43% or 70% survival in animals treated with 7.5, 15 or 30 U·kg^−1^ of Anthrasil. The survival induced by Anthrasil is statistically significant, but there is no statistical difference between each dose of Anthrasil.

#### 2.2.2. Antibiotherapy

Antibiotherapy is the reference treatment for anthrax. All types of anthrax infections can be treated with antibiotics. Even though antibiotics cannot neutralize the toxins, they are effective in overcoming bacteremia caused by antibiotic-susceptible strains of anthrax. The CDC has developed guidelines for the prevention and treatment of naturally occurring or bioterrorism-related anthrax in conventional medical settings [51,52,53]. However, during an anthrax mass casualty incident, resource limitations might warrant a shift to contingency or crisis standards of care. In addition, antibiotic shortages have been described in some countries (e.g., Amoxicillin shortage in France and the United Kingdom in the winter of 2022), which may complicate treatment. Such shortages strengthen the necessity of building strategic drug stockpiles.

Anthrax spores typically take one to seven days to germinate and secrete toxins, but some of them can germinate after sixty days. Due to this long germination time, antibiotherapies have to be administered for 60 days. Conventional treatment is based on Ciprofloxacin and Doxycycline, which offer the same protection against anthrax. Although both antibiotics have some potential, and serious, side effects, the expected benefit outweighs these risks. Adult treatment requires 100 mg of Doxycycline or 500 mg of Ciprofloxacin, twice a day for 60 days. For anthrax treatment, the CDC has also approved Levofloxacin, Penicillin G and Amoxicillin for women who are pregnant or breastfeeding and children.

During the anthrax attack in the USA, antibiotic prophylaxis was administered to ∼32,000 individuals suspected of having been exposed to anthrax [54]. Nevertheless, poor patient compliance was observed. As a result of the attack, at least eleven people were contaminated, five of whom died despite the rapid administration of treatment (antibiotics and intensive care). This observation is consistent with the conclusion of previous analyses, showing that the course of anthrax may progress to a point where the levels of secreted anthrax toxins are such that death occurred despite the administration of antibiotics.

Although natural resistance to antibiotics has only rarely been documented for *B. anthracis*, in vitro studies have shown that *B. anthracis* can develop resistance to Ciprofloxacin, Doxycycline, β-lactam and Fluoroquinolone antibiotics [55]. It was also demonstrated that an efflux pump encoded by the *gyrA*, *gyrB* and *parC* genes can mediate cross-resistance to Fluoroquinolone antibiotics like Ciprofloxacin in *B. anthracis* [56]. It was also demonstrated that *B. anthracis* expresses the *bla1* and *bla2* genes that are capable of expressing beta-lactamases [57]. The development of safer and more effective chemotherapeutic options is necessary.

#### 2.2.3. Chemical Inhibitors

Studies have demonstrated the activity of novel compounds extracted from medicinal plants [58]. Some phytochemical compounds are able to interact synergistically with antibiotics that are already available, which would be helpful in preventing the emergence of antibiotic resistance. With a synergistic effect, it would be possible to administer lower doses of antibiotics, which could reduce their side effects. Nevertheless, the data are limited, and therefore, broader studies are needed. Alkaloids are plant-derived compounds that can intercalate with bacterial DNA and inhibit enzymes associated with nucleic acids, such as esterase or DNA or RNA polymerases [59]. Tomatidine has been demonstrated to be effective on *B. anthracis*, by inhibiting ATP synthase activity [60]. Terpenes disrupt the bacterial cell membrane due to their lipophilic nature. Quinones may target bacterial peptidoglycan and enzymes associated with the cell membrane. Anthraquinone has demonstrated potency against *B. anthracis*. Such plant-derived molecules could be useful for the treatment of resistant strains.

#### 2.2.4. Vaccine

Anthrax vaccine adsorbed (AVA, BioThrax) is a subunit vaccine approved by the FDA (gaining formal FDA approval in 2008) for persons at high risk of exposure and/or potentially exposed to anthrax [61]. AVA is produced from microaerophilic culture filtrates of a toxigenic, but avirulent, non-encapsulated mutant V770-NP1-R of the *B. anthracis* Vollum strain and is composed chiefly of PA with small amounts of LF and EF that may vary from batch to batch. The vaccine is adjuvanted with aluminum hydroxide. Five shots (IM) of AVA are required to complete the vaccination schedule.

BioThrax can be used as pre-exposure prophylaxis and as post-exposure therapy. As a prophylactic, three initial doses of the vaccine are required (at 0, 1 and 6 months). Then, a booster series is required (at 6 and 12 months, and then 1 dose every 12 months). As post-exposure therapy, unvaccinated people over the age of 18 who have been exposed to aerosolized *B. anthracis* spores must receive three doses of BioThrax (at 0, 2 and 4 weeks) combined with antimicrobial therapy.

A five-year safety study was carried out on 7000 workers at high risk, who received almost 16,000 doses of BioThrax administered by the subcutaneous route. Severe, moderate and mild local adverse reactions were observed in 0.15%, 0.94% and 8.63% of patients, respectively. Four cases of transient systemic adverse reactions were reported (<0.06% of doses administered).

The US federal government has a goal of stockpiling 75 million doses of BioThrax, as part of the BioShield Act project. In 2019, more than 8 million doses of BioThrax were administered to 1.9 million people.

A third-generation vaccine candidate is under development. [62,63,64] (NuThrax) could be considered a new formulation of BioThrax. AV7909 is composed of the AVA drug substance and the adjuvant CPG 7909, an immunostimulatory Toll-like receptor 9 (TLR9) agonist that enhances antigen-specific antibody response and natural killer T-cell response. AV7909 was shown to increase the magnitude of immune response, thereby shortening the time to protective immunity. A safety study was conducted in Sprague Dawley rats. The rats received three intramuscular doses (IM) of AV7909 on days 1, 15 and 29 [63]. The animals were observed for two weeks after the termination of the treatment to evaluate the persistence or reversibility of any toxic effects. In the adult rat model, the maximal safe dose is 0.5 mL AVA with 0.5 mg of CPG 7909. The AV7909 vaccine was demonstrated to be safe and well tolerated. Guinea pigs and non-human primates were exposed to an aerosol challenge of 200 LD_50_ of *B. anthracis* Ames strain spores. All untreated control guinea pigs and control non-human primates died from anthrax. In contrast, AV7909 vaccination conferred protection in a dose-dependent manner in both the day 28 and day 70 challenge cohorts.

## 3. Ricin

### 3.1. Background

Ricin is classified in the CDC’s category B and is part of the List 1 of the Chemical Weapons Convention (CWC, schedule 1 compound) and the Biological Weapons Convention (BWC). Nevertheless, ricin is one of the major potential biowarfare agents. The ricin toxin is produced in the endosperm of the seeds of *Ricinus communis* (L. Euphorbiaceae family), also known as castor bean. The plant is present worldwide as an invasive species or cultivated as an ornamental plant or a plant used for industry [65]. There are two isoforms of ricin: ricin D and E. Structurally, ricin is a disulfide-linked heterodimeric glycoprotein (an AB toxin) consisting of the ricin toxin A (RTA) chain and the ricin toxin-binding B (RTB) chain. RTA functions as an N-glycosidase, which cleaves adenine 4324 from the 28S ribosomal RNA within the 60S ribosomal subunit. The cleavage irreversibly inactivates the ribosome and stops the protein synthesis. RTB is a galactose-specific lectin that binds to glycolipids and glycoproteins on the surface of vertebrate cells and facilitates the translocation of RTA into the cytosol. Remarkably, a single molecule of ricin can deactivate 1000 to 1500 ribosomes per minute. The human lethal dose (LD50) of ricin varies by exposure route, estimated between 2 and 10 µg·kg^−1^ via aerosol or parenteral routes, and 1–20 mg·kg^−1^ via ingestion [66,67,68].

Ricin continues to be produced in large quantities for industrial and pharmaceutical purposes. In 2020, 2050 tons were produced despite a decline in the use of ricin-derived products in more developed countries.

Several potential uses of ricin as a biological weapon have been documented. During World War II, ricin was weaponized by the USA and the United Kingdom under the code name “compound W”. The most notorious case of ricin being used as a bioweapon was the assassination of Bulgarian dissident Georgi Markov in London in 1978. Markov was shot with a micro-engineered pellet, potentially containing ricin, by an umbrella wielded by an operative associated with the Bulgarian Secret Service. During a United Nations inspection in Iraq in 1990, following the Gulf War, it was discovered that Iraq had possibly weaponized 10 L of ricin. In 2001, the “al-Qaeda Chemist” was arrested for having produced ricin. In 2003, vials containing ricin traces were found in a French train station. Letters and parcels containing ricin were sent to Senator Bill Frist, President Obama, President Trump and the Pentagon in 2004, 2013 and 2018, respectively. In 2018, a Lebanese individual linked to Daesh was arrested in Italy, suspected of planning to contaminate water supplies with ricin or anthrax. In the same year, police in Cologne arrested a man affiliated with Daesh, seizing 84.3 mg of ricin and approximately 3300 ricin seeds, along with explosive devices containing ricin. This history of ricin use underscores the critical need for prophylactic measures and therapeutic options.

Ricin poisoning symptoms are generally consistent regardless of the exposure route, though some symptoms are more specific. The severity of symptoms is dose-dependent. Typically, symptoms appear 2 to 6 h after ingestion and within 8 h after inhalation, although they can sometimes be delayed by up to 20 h. Physical symptoms include abdominal pain, vomiting, diarrhea (with or without blood), muscle pain, limb cramps, circulatory collapse, breathing difficulties and dehydration. Muscle pain and circulatory collapse are more frequently associated with injected ricin, along with pain at the injection site. Exposure to ricin aerosol can cause skin and eye irritation, respiratory distress, fever, cough, nausea, chest tightness, heavy sweating and fluid accumulation in the lungs (pulmonary edema). Biochemical analyses often reveal elevated white blood cell counts and blood urea nitrogen, aspartate aminotransferase and alanine aminotransferase levels, indicating liver and kidney dysfunction. Autopsies of peoples who died from ricin poisoning revealed hemorrhagic necrosis in the intestines and heart, and edema in the lungs.

### 3.2. Therapy

Currently there are no approved therapies for the specific treatment of ricin intoxication. Therapy is only based on the treatment of individual symptoms. The development of specific therapies is essential.

#### 3.2.1. Antibodies

The efficacy of antibody inactivation of the ricin toxin has been recognized for more than a century. The epitope map of ricin holotoxin has been constructed and the hot zones for neutralization have been identified [69]. Several murine or recombinant antibodies neutralize ricin activity in vitro and protect mice from death in in vivo challenges. Anti-ricin antibodies can block ricin’s entry into cells, but they can also hinder its intracellular routing.

Yu et al. isolated human antibodies (from transgenic mice) that neutralize ricin [69]. The 4–4E human antibody (IgG1) targets RTA and has a half-maximal effective concentration (EC_50_) of 22.58 µg·mL^−1^ [70]. An amount of 2.5 mg·kg^−1^ of 4–4E, administered up to 24 h before an IP challenge with 2 mLD_50_ of ricin, fully protected mice from death. Yu et al. have also developed a 4–4E DNA-encoded monoclonal antibody (DMAb). Using intramuscular electroporation (IM EP), the 4–4E DMAb is rapidly expressed in vivo within seven days and is enriched, both in the intestines, and, mostly, in the gastrocnemius muscle. Five days after electroporation, all the mice that received 4–4E DMAb survived the IP challenge with 2 mLD_50_ of ricin. The 4–4E could interfere intracellularly with the protein disulfide isomerase (PDI)-mediated reduction of ricin.

huPB10 (hPB10) is a humanized antibody that targets the enzymatic subunit and neutralizes ricin by interfering with its transport to the trans-Golgi complex network [71]. Its affinity is similar to that of the parental antibody PB10 = 40 nM. An IP administration of 5 or 40 µg of huPB10, 24 h pre-intoxication, fully protected mice from death after an IP or intranasal challenge with 10 LD_50_ of ricin. Rong et al. also performed a protection assay on mice intoxicated with ricin via the pulmonary route. Intranasal administration of an oligoclonal mixture composed of huPB10 and huSylH3, another humanized antibody, nullified the effects of the mice’s pulmonary ricin infection. They also demonstrated that the coadministration of immune complexes composed of ricin and antibodies, via the intranasal route, induces protective immunity. Indeed, all mice that received these immune complexes survived a lethal challenge with 5 LD_50_ of ricin 90 days later [72].

43RCA-G1 is a humanized recombinant antibody derived from macaques (humanized variable domains grafted on human IgG1 constant domains). 43RCA-G1 binds the ricin A chain with an affinity of 40 pM [73]. In a cell-free neutralizing assay, 43RCA-G1 neutralized 89% of ricin activity at 40 µg·mL^−1^, and 50% at 1.5 µg·mL^−1^. 43RCA-G1 also protects cynomolgus monkeys from ricin (24 μg·kg^−1^) intoxication (intranasal administration) when the antibody is nebulized [74]. 43RCA-G1 was also tested as an oligoclonal mixture, with RB34, a murine antibody. The oligoclonal mixture neutralizes ricin, with an IC_50_ 2 to 12 times lower than that of each Ab individually (*p* < 0.01), except for two ricin cultivars [75].

Polyclonal antibodies may be an alternative to recombinant and monoclonal antibodies. A F(ab’)2-based anti-ricin antitoxin was produced from the hyperimmune plasma obtained from a vaccinated horse [76]. Swine were challenged with crude ricin at a dose of 3 µg·mL^−1^·kg^−1^ (intratracheal route) or 7.5 mg·kg^−1^ (intramuscular route). Injection of the antitoxin 18 h after the challenge protected more than 80% of both intratracheally and intramuscularly ricin-intoxicated swine. Injection of the antitoxin 24 h after the challenge protected 58% of the intramuscularly exposed swine as opposed to 26% of the intratracheally exposed animals.

JJX12 is a bispecific antibody consisting of RTA-D10, a camelid single variable domain (V_H_H) antibody targeting RTA, linked to the V_H_H RTB-B7, that targets RTB [77]. The protection induced by JJX12 is better than the protection conferred by an equimolar mixture of both parental antibodies. When JJX12 and ricin were incubated together for 1 h, with an antibody/ricin ratio as low as 4:1, all mice survived.

#### 3.2.2. Chemical Inhibitors

There are currently no chemical inhibitors approved for the treatment of ricin intoxication, but several molecules are promising.

VPg1–110 is a molecule composed of an N-terminal truncated variant of a viral genome-linked protein (VPg) from a turnip mosaic virus (TuMV). VPg1-110 binds to RTA and inhibits the depurination of 28S rRNA in vitro [78].

Retro-2 is a small molecule that neutralizes ricin. Retro-2 induces ricin accumulation in early endosomes and the relocalization of the Golgi SNARE protein syntaxin-5 to the endoplasmic reticulum [79]. The advantage of this molecule is that it neutralizes the ricin intracellular pathway instead of targeting the toxin itself. Because Retro-2 targets a ubiquitous pathway, it also neutralizes various pathogens such as toxins, viruses, intracellular parasites and bacteria. Retro-2 has been assayed in vivo. Mice received an intraperitoneal (IP) dose of Retro-2, before a challenge with an intranasal dose of ricin leading to 90% mortality at day 21. A dose of 200 mg·kg^−1^ of Retro-2 fully protected the mice [80]. Retro-2 was solubilized in DMSO, and no significant toxicity was observed for animals after intraperitoneal administration up to 400 mg·kg^−1^. Retro-2 and its derivative, Retro-2.1, represent a drug of choice for further clinical development.

EACC (ethyl(2-(5-nitrothiophene-2-carboxamido)thiophene-3-carbonyl)carbamate) also neutralizes ricin and other plant toxins in vitro. EACC seems to inhibit the release of the ricin A-chain in the endoplasmic reticulum and is possibly also coupled with a decrease in translocation from the endoplasmic reticulum to the cytosol [81].

#### 3.2.3. Vaccines

Ricin is a plant toxin. As a noninfectious pathogen and a rare disease, the balance between the risk and the benefit of global vaccination is not favorable. A ricin vaccine would only be useful for people at high risk of exposure, such as soldiers or first responders.

Previous studies have shown that the A chain is more immunogenic than the B chain, but both chains can be used for vaccination [82]. Two recombinant vaccines based on immunization with the RTA subunit are currently in Phase 1 and 1b development, namely RVEcTM (USAMRIID, United States Army Medical Research Institute of Infectious Diseases, Fort Detrick, MD, USA) and RiVax^®^ (University of Texas Southwestern Medical Center, Dallas, TX, USA) [83].

RiVax^®^ is a subunit vaccine under development that is efficient with any route of contamination. The ThermoVax^®^ technology was used to enhance the stability of RiVax^®^. The vaccine is stable at least 40 degrees Celsius for up to one year. Such stability is a major advantage for stockpiling and for shipment all over the world in a military context. RiVax^®^ is composed of a ricin A-chain that has been modified by the addition of two mutations removing its toxic activity. The mutations, Y80A and V76M, disrupt the ribotoxic site and the vascular leak syndrome-inducing site, respectively [84,85]. RiVax^®^ is adjuvanted with alum and can be administered via the intramuscular route. Three doses are required for an optimal neutralizing response. RiVax^®^ has been shown to be safe in two phase 1 studies in humans, and its development has been continued under the Animal Rule. Rhesus macaques received three doses of RiVax^®^ before being challenged 110 days later with ~5 LD_50_ of aerosolized ricin. All vaccinated animals developed ricin-specific antibodies, and the majority of the animals developed neutralizing antibodies. A correlation has been observed between vaccine-induced serum antibody profiles derived from a competitive ELISA, and survival following exposure to a lethal dose of RT in an NHP model [86].

A recombinant ricin vaccine from *E. coli* (RVEc™) was assessed in a phase 1a clinical trial. RVEcTM is composed of an inactive truncated ricin A-chain that lacks the hydrophobic carboxy-terminal region (residues 199–267) as well as a small hydrophobic loop in the N-terminus (residues 34–43), resulting in a molecule with increased solubility and thermal stability [87,88,89]. RVEcTM is adjuvanted with Alhydrogel^®^, so the development of an o-phthalaldehyde (OPA) assay was realized to measure the protein content in an *E. coli* ricin vaccine [90]. During the clinical Phase 1a, RVEc™ was safe and well tolerated for all doses tested.

## 4. Botulism

### 4.1. Background

Botulinum neurotoxins (BoNTs) are the most lethal biological substances for humans in the world. Their LD50 is estimated up to 1 ng·kg^−1^ [91]. Natural intoxication occurs via ingestion of BoNTs or *Clostridium Botulinum* (*C. Botulinum*) spores. In the context of terrorism, intoxication via inhalation or injection is probable. Natural botulism is rare, but still lethal. In September 2023, in France, a botulism outbreak contaminated fifteen people from eight countries, and of the ten people who were hospitalized as a result, one died.

BoNTs have been classified as Tier 1 biothreat agents by the United States CDC, since a single gram would theoretically kill more than a million people if it were dispersed effectively. Several bioterrorism scenarios have flagged the risk of the contamination of the food chain by the intentional spread of BoNTs [92].

The development of botulinum toxins as a bioweapon began some time before World War II, when Chinese prisoners were fed cultures of *C. botulinum* during the Japanese occupation of Manchuria in 1931 [93,94]. During World War II, the US biological weapons program first produced BoNTs because of concerns that Germany had weaponized BoNTs. As a result, more than one million doses of the BoNT toxoid vaccine were prepared for Allied troops before the invasion of Normandy on D-Day [95]. Although the Biological and Toxin Weapons Convention (1972) prohibited the development of biological weapons, Iraq and the Soviet Union produced botulinum toxin as a weapon. BoNT was one of several agents tested at the Soviet site Aralsk-7 in the Aral Sea. A former senior scientist of the Russian bioweapon program reported that Soviets attempted to splice the BoNT gene into other bacteria. After the fall of the Soviet Union, the thousands of scientists employed in bioweapons programs were recruited by nations attempting to develop biological weapons [96,97]. BoNTs were also used by terrorists as a bioweapon when aerosols were dispersed in downtown Tokyo and at US military installations in Japan on several occasions by the Japanese cult Aum Shinrikyō between 1990 and 1995. Luckily, these bioterror attacks failed due to faulty microbiological techniques and inefficient aerosol dispersion. (WuDun 1998 New York Times). Iran, Iraq, North Korea and Syria are suspected of having developed BoNTs as a bioweapon. At the end of the Gulf War in 1991, Iraq admitted to the United Nations that it had produced 19,000 L of concentrated BoNT, including 10,000 L already loaded into military weapons. A volume of 19,000 L of BoNTs represents about three times the amount needed to kill the entire human population via inhalation, knowing that far less is needed via injection [98]. In 1990, Iraq deployed thirteen missiles filled with botulinum toxin, ten with Aflatoxin and two with anthrax spores, as well as bombs for immediate use. This comprised 100 bombs containing BoNTs, 50 containing anthrax spores, and 7 containing aflatoxin. The deliberate release of botulinum toxins into a civilian population can cause massive disruption and distress. In addition, terrorist use of botulinum toxins might be perpetrated via the deliberate contamination of food or water supply.

BoNTs are produced by bacteria of the genus *Clostridium* which are Gram-positive, anaerobic spore-forming microorganisms. They are mainly produced by *C. botulinum* but also by atypical strains of other *Clostridium* spp., such as *Clostridium butyricum* and *Clostridium baratii*. Some strains of *C. sporogens* also produce BoNTs [99]. BoNTs constitute a vast growing family of >40 toxin variants grouped into 10 serotypes (toxinotypes): A, B, C, D, E, F, G, H (also referred as H/A or F/A), X and J, and more than 40 subtypes [100,101,102,103]. Serotypes A, B, E and to a lesser extent F can naturally cause human botulism [104]. BoNT/F is extremely rare and often associated with infant botulism [104]. Serotypes C and D are generally involved in animal botulism. Even if natural infections are limited to certain serotypes, all of them are infectious to humans and could be used for bioterrorism. BoNTs have been intentionally used as bioweapons in military conflicts or terrorist attacks [105]. The Working Group on Civilian Biodefense in the USA has assessed this risk [105]. The rapid progress in genomic information has revealed the presence of BoNT-related sequences , such as the BoNT/X found in a new gene cluster of a *C. botulinum* strain 111, and in non-clostridial strains the BoNT/Wo or BoNT/I detected in the genome of the *Weissella oryzae*, a bacterium isolated from fermented rice, the BoNT/J (ebont/F or BoNT/En) found in the genome of a strain of *Enterococcus faecalis* isolated from cow feces, and Cp1 from *Chryseobacterium piperi* isolated from sediment [106,107,108]. However, the public health implications of the presence of BoNT genes in non-clostridial strains on human or animal botulism remain to be defined [100].

Botulism occurs in the form of four different clinical syndromes: foodborne, wound, infant botulism and adult intestinal toxemia. Moreover, inhalational botulism results from the aerosolization of BoNTs in rare cases of laboratory botulism, and iatrogenic botulism can result from the injection of BoNT overdoses after therapeutic or cosmetic use [100,106,107]. The botulism clinical syndrome itself is a paralytic illness starting with symmetrical cranial nerve palsies followed by the descending, symmetric flaccid paralysis of voluntary muscles, which may progress to respiratory compromise and death. The treatment includes a long stay in an intensive care unit, with mechanical ventilation when necessary, and the administration of an antitoxin as soon as the diagnosis is confirmed. Moreover, *C. botulinum* spores might lead to a toxico-infection by colonization of the intestinal tract and in situ BoNT production. Children who are under one-year old can develop infant botulism, since they are more susceptible to intestinal colonization by *C. botulinum* [109,110]. Wound botulism is a consequence of contamination with *C. botulinum* spores leading to in situ growth and BoNT production.

BoNTs are produced as large protein complexes combining a neurotoxic subunit with a nontoxic non-hemagglutinin (NTNH) component, and with either hemagglutinin (HA) or OrfX proteins. BoNTs are composed of a light chain (LC, 50 kDa) and a heavy chain (H, 100 kDa), linked by a disulfide bond [111,112]. The C-terminal domain of HC binds to polysialic gangliosides and protein receptors on neuronal membranes [113], triggering the BoNT internalization by endocytosis [114]. The N-terminal domain of the H chain (HN) contributes to the translocation of LC into the cytosol. The LC is a zinc-metalloprotease that cleaves one of the three SNARE proteins (SNAP-25, VAMP and syntaxin) involved in neurotransmitter exocytosis [115].

### 4.2. Therapy

#### 4.2.1. Antibodies

The only proven and specific therapy post-BoNT intoxication is the administration of a BoNT serotype-specific antitoxin [115]. However, BoNT polyclonal antibodies are mainly of equine origin [116]. The trivalent equine-derived antitoxin became available in the 1960s and has been widely used [116], but this current antitoxin (BAT) is a Fab’2 polyclonal antibody that presents the risk of hypersensitivity reactions, including cardiac arrest and serum sickness. This trivalent formulation was withdrawn from the market in 2018. Currently, no licensed BoNT-specific antitoxin is available in the European Union. In addition, this antitoxin did not cover the BoNT serotypes C, D, F, G, HA, X or J. This antitoxin shortage has been temporarily resolved by imports of limited doses of a heptavalent BoNT antitoxin produced in the United States exploiting special permission, but long-term supply beyond 2022 has not been ensured, as there is no EMA market authorization and the heptavalent product does not cover new BoNT types. Therefore, it is of the utmost importance to focus efforts on research into the independent EU production of a therapeutic product for the treatment of BoNT intoxication to protect the civilian population of Europe against the deliberate release of any BoNTs.

In October 2003, the US Food and Drug Administration approved a human botulinum immune globulin derived from the pooled plasma of human adults immunized with pentavalent botulinum toxoid, which is currently used to treat infant botulism via the intravenous route (BIG-IV, babyBIG^®^) [117]. This blood product bears a far lower risk of anaphylaxis compared to trivalent equine antitoxin [118]. The BIG-IV immune globulin is produced by plasmaphersing laboratory personnel immunized with pentavalent botulinum toxoid due to the risk of exposure to BoNTs, but its production is not scalable.

As of 13 March 2010, BAT^®^ (Botulism Antitoxin Heptavalent, Cangene Corporation, Winniped, MB, Canada) replaced all other non-infant botulinum antitoxins. BAT^®^ is composed of fragments of IgG directed against seven BoNT types (A to G) and is derived from equine plasma consisting of more than 90% Fab or F(ab’)2 immunoglobulin (Ig) fragments to reduce hypersensitivity reactions. The BAT^®^ treatment may require several injections, since the Fab and F(ab’)2 fractions are eliminated from blood circulation more rapidly than intact IgGs [119]. For example, this occurred in a patient with BoNT/F intestinal botulism showing improvement after a single administration of HBAT, but when Fab/F(ab’)2 IgG fragments were cleared from circulation, the BoNT/F rebounded, and bilateral descending flaccid paralysis recurred [120]. For the protection of European civilians against the deliberate release of any BoNTs, research into a therapeutic product for the treatment of BoNT intoxication is paramount. Indeed, it was recently found that recombinant BoNT subunits may replace botulinum toxoids as more efficient and safer antigens for the preparation of pharmaceutical anti-botulinum equine antitoxins [121]. Hyperimmune antisera from large mammals, in particular horses, are routinely used for life-saving anti-intoxication intervention, but are complicated by their possible reactogenicity and limited availability. There is an urgent need for alternative safer next-generation immunotherapies. The development of new tools for equine antibody engineering allowed the generation of immune phage display libraries, representing highly diverse V-gene repertoires of horses immunized against botulinum A or B neurotoxins. Highly specific scFv clones carrying equine V-genes and human Gamma1/Lambda constant genes have been referred to as “Centaur antibodies”. Preliminary assessment in a murine model of botulism established their therapeutic potential as a valuable tool for engineering safer therapeutic equine antibodies [122].

Several monoclonal and recombinant antibodies are under development. They offer an unlimited source of selective agents that do not present a risk of anaphylaxis or serum sickness. Several mouse mAbs against BoNT/A, B, E and F, have been well characterized in the form of unique mAb formulations or as a combination of three mAbs [123,124,125,126]. Highly specific mAbs against the three main botulinum toxinotypes were generated by immunizing mice with a mixture of HC/A, B and E. This technique led to a synergistic effect of an up to 400-fold enhancement in neutralizing power against BoNT/A, B and E in a formulation that is potentially useful in diagnosis and therapy [127]. The creation of this type of triplex antibody, able to neutralize large doses of BoNTs, represents a significant improvement and a safe substitute for polyclonal sera [128,129]. A combination of two human mAbs was able to completely neutralize BoNT/B1 and was effective both prophylactically and therapeutically in the mouse bioassay. This type of human mAb combination offers a broad neutralizing profile against a range of B subtypes [130]. A mixture of mouse mAbs against BoNT/A, B, E and F was generated, which neutralized the highest dose of BoNT/A in vivo [131]. Six highly protective sheep mAbs (SmAbs) were produced using a BoNT/A1 toxoid or BoNT/A1 HCc for immunization. The resultant SmAb combinations were highly protective with the trivalent combination offering 100% protection against botulism [132]. A panel of camelid antibodies directed against the light chain of BoNT/E offered insight into BoNT/E function and served as the basis to develop antidotes following BoNT/E exposure [133]. An equimolar mixture of three human IgG monoclonal antibodies (NTM-1633), targeting BoNT serotypes E1, E2 and E3 was assessed in a double-blind, single-center, placebo-controlled dose escalation study randomized in three cohorts of healthy volunteers receiving a single intravenous dose of NTM-1633 (0.033, 0.165, or 0.330 mg/kg) or saline placebo. NTM-1633 had a favorable PK profile with a half-life >10 days and a favorable safety and immunogenicity profile, supporting its development as a treatment for BoNT/E intoxication and postexposure prophylaxis [134].

According to Chen et al., the determination of key epitopes is paramount to developing effective antibodies. They showed that the combination of two mAbs recognizing different receptor binding sites had a synergistic effect against BoNT/Bs. A combination of 3 mAbs directed against BoNT/A or BoNT/B prevented botulism after an aerosol challenge with BoNT/A1 or BoNT/B1, showing that mAb combinations represent an alternative to vaccination in the case of mass exposure to BoNTs [135]. A single IM injection of the combination administered 48 h pre-exposure protected guinea pigs against an aerosol challenge of BoNT/A1 and BoNT/B1, providing pre-exposure prophylaxis against botulism from aerosol exposure [136,137]. Additionally, a bispecific antibody, LUZ-A1-A3, was constructed, demonstrating effective and potent neutralization of BoNT/A by binding to the AHc and AL-HN domains. In vivo experiments in mice revealed that LUZ-A1-A3 could serve as a potent prophylactic and therapeutic agent against BoNT/A as a bispecific antibody [138].

Existing mAbs can also be exploited by selecting and shuffling VH and VL domains from a variety of repertoires to build highly effective chimeric antibodies. These shuffling chimeric antibodies represent a suitable candidate to replace horse antitoxins and avoid the risk of serum sickness [139]. A similar approach was used to generate a powerful chimeric antibody against BoNT/A, namely TA12 [69]. Brier et al. evidenced the epitopes and the influence of the gangliosides in TA12 binding. Hence, TA12 potently blocks the entry of BoNT/A1 into neurons by interfering simultaneously with the binding of SV2C and, to a lesser extent, GT1b. This study reveals the unique neutralization mechanism of TA12 and emphasizes the potential of using single mAbs for the treatment of botulism type A [140]

In 1997, Amersdorfer et al. utilized phage antibody libraries to generate antibodies capable of neutralizing BoNT/A following mouse immunization with BoNT/A HC [141]. They also studied the immune response to BoNT/A binding domains using phage display on scFv isolated from a human volunteer immunized with a pentavalent botulinum toxoid. Their findings revealed that the pentavalent botulinum toxoid directs the humoral immune response to a limited number of epitopes on the HC binding domain [142]. Two specific neutralizing antibodies targeting different epitopes were selected from a fully synthetic human antibody library using purified BoNT/A HC fragments, demonstrating the effectiveness of fully synthetic human antibody phage display libraries in providing therapeutic antibodies [143]. The use of yeast display in combination with noncanonical amino acids (ncAAs) can provide irreversible variants of single-domain antibodies (sdAbs), called VHHs and nanobodies, targeting botulinum neurotoxin light chain A [144]. Using molecular techniques, it has become possible to both increase the affinity and maintain cross-toxinotype reactivity for the antibodies that potently neutralize BoNT in vivo [123]. Yeast-displayed mutants were also used to obtain a three-mAb combination potently neutralizing BoNT/FA or HA that could serve as prevention and treatment of type FA or HA botulism [145]. Yeast-displayed scFv libraries were prepared from mice immunized with BoNT/G and screened using fluorescence-activated cell sorting (FACS). Three IgG combinations completely protected mice challenged with 10,000 LD50s of BoNT/G. Such mAb combinations have the potential, in combination with antibodies to BoNT/A, B, C, D, E and F, to provide the basis for a fully recombinant heptavalent botulinum antitoxin to replace the currently only available equine product [146].

Oligoclonal antibodies were isolated from libraries obtained from humans immunized with BoNT/A, B, C, D and E toxoids [147]. The oligoclonal approach is relevant for botulism, because the oligoclonal mixture can be administered without serotyping as a prerequisite. NTM-1633 is an oligoclonal antibody (three antibodies) in pre-clinical development for the neutralization of BoNT/E_1_, E_3_ and E_4_ [148]. The XOMA3AB is another oligoclonal antibody (three antibodies), but one that neutralizes BoNT [149]. Due to the very limited availability of these antitoxins outside the USA, the development of countermeasures in Europe is urgent. The objective of the AntiBotABE program was to develop recombinant antibodies neutralizing BoNT/A, B and E. To be well tolerated, immune libraries were generated after the immunization of *Macaca fascicularis*, and the antibodies were germline-humanized. Oligoclonal antibodies (two antibodies by serotype) protect mice from challenges with BoNT/A and BoNT/B. The antibody hu8ELC18 protected mice from death after a challenge with BoNT/E. This project confirmed the potential of expanding the therapeutic spectrum while decreasing the number of antibodies in a broad-spectrum mixture [150].

Despite the development of recombinant antibodies, polyclonal antibodies still represent an alternative to rapidly obtain a medical countermeasure. The medical treatment of botulism has historically consisted of antibody therapy using formalin-detoxified toxins and, more recently, recombinant or chemically altered derivatives of the toxins [151,152]. In the Keller Laboratory, formalin-detoxified toxoids were prepared from BoNT/A using optimized formaldehyde reaction conditions. The protective IgG titers induced by formalin toxoids vary greatly depending of the quality of the toxoids [153].

Polyclonal antibodies were also developed starting from horses immunized with monovalent toxins inactivated with formalin [154]. These antitoxins have been used in Japan for more than 50 years. A recent clinical study on a total of 134 patients suggested that the antitoxin therapy in Japan is safe and highly effective. Nevertheless, more efforts are still needed to develop antitoxin formulations with a scalable production process and a better safety profile [155].

Effective antibodies can be isolated from dromedary-derived libraries. Nanobodies (also referred to as variable domains of heavy-chain antibodies, VHH) can be produced with a high yield and are stable, which is interesting for the industrial production. Bakherad et al. identified a VHH that fully protects mice from death (150 ng·kg^−1^) after a challenge with 3LD_50_ of BoNT/E [156].

#### 4.2.2. Antibiotherapy

Botulism is caused by bacteria, and mainly by *C. botulinum*. Nevertheless, contamination generally occurs by eating food contaminated by the spores/the toxin. In this kind of intoxication, antibiotics are not effective. Antibiotherapy, mainly based on Penicillin G and Metronidazole, may only be used, in combination with antitoxins, for the treatment of wound botulism. Antibiotics are also recommended for the treatment of secondary infections in infant botulism. Treatment with aminoglycoside antibiotics may potentiate the effect of botulinum neurotoxin at the neuromuscular junction, even in infants that have been treated with BabyBIG^®^. Therefore, extra caution should be exercised when weaning ventilatory support from infants who have infant botulism and have received aminoglycoside antibiotics. Indeed, antibiotics do exacerbate the release of pre-formed BoNTs after bacterial lysis.

#### 4.2.3. Chemical Inhibitors

Currently, there are no approved small molecules. Nevertheless, increased efforts are required, since small inhibitors are thought to prevent the neuroparalytic action of BoNTs, irrespective of their serotype or subtype. None of the reported small molecule agents have shown therapeutic utility or protection in mice due to limited efficacy, poor cell membrane permeability or cytotoxicity. Some of them have induced an increase in mean time to death, however, more small molecule inhibitors, with suitable absorptions, distributions and safety profiles, are still urgently needed [157]. Recently, one small inhibitor (EGA) was shown to prevent the neurotoxicity of various BoNTs by interfering with their entry route, reducing botulism symptoms in mice. This opens up the possibility of using EGA as a good candidate for the development of new inhibitors [158]. The aminopyridine antagonists (DAPs) of voltage-gated potassium channels have recently been evaluated for the treatment of botulism. However, the mechanisms responsible for the reversal of paralysis are still poorly understood and further clinical studies are needed to confirm the efficacy of DAP as a botulism treatment [159]. More recently, post-symptomatic treatment with nontoxic BoNT derivatives reversed botulism symptoms and saved mice, guinea pigs and non-human primates after injection with a lethal dose of BoNT/A1. This seminal work established for the first time that therapeutic proteins can be delivered into the neuronal cytoplasm, where they reverse intoxication by BoNTs [160]. The same authors established that neurotransmission was significantly improved at 21 days post-intoxication in 3,4-DAP-infused rats, although it was still depressed compared to naïve rats. They showed that 3,4-DAP is the first small molecule to reverse systemic paralysis in animal models of botulism, thereby meeting a critical treatment need that is not addressed with conventional antitoxins [161].

More recently, a novel tetrapeptide RRGW has been found to significantly delay BoNT/A-induced muscle paralysis in the DAS (digit abduction score) assay in mice. This peptide inhibitor is a promising drug candidate for the treatment BoNT/A poisoning [162].

#### 4.2.4. Vaccines

BoNTs are closely related to tetanus toxins (TeNTs), and vaccination based on tetanus toxoid is a universal strategy for tetanus prevention. However, BoNTs have both toxic effects and therapeutic benefits, so vaccination is not desirable as a widely applied measure. For example, the most well-known aesthetic and therapeutic preparation, BOTOX^®^ (OnabotulinumtoxinA), is approved for eleven therapeutic indications [163,164]. Vaccines against BoNTs are scarcely used, since botulism is a rare disease, and vaccination would hinder the use of medicinal preparations of botulinum toxins [119]. Vaccination is restricted to individuals with a high risk of exposure to BoNTs, such as healthcare providers, researchers, first responders and military personnel [165]. The pentavalent botulinum ABCDE toxoid vaccine was administered to military personnel and workers at risk of exposure to BoNTs, but this method involved high production costs and could be dangerous during detoxification. Thus, the CDC discontinued the botulinum vaccine program for workers at risk for occupational exposure to BoNTs in 2011. In addition, minute doses of this deadly poison are used therapeutically to locally paralyze muscles for clinical or cosmetic benefits [166,167]. BoNTs have been approved for dystonia, blepharospasm, chronic pain, migraines, overactive bladder problems and inflammation [168], and recent clinical studies describe their potential application in major depressive disorders [164,169].

Since antibodies only prevent further exposure to the toxin, vaccination is essential for populations at risk of exposure [170]. Plasmids containing genes encoding for BoNT-HCs, replicon DNA or particle pentavalent vaccines could simultaneously induce antibody responses and protect against five agents including BoNT/A, B, E and F, offering the opportunity to produce large and nontoxic quantities of BoNT domains or subunits [171]. Other suitable vaccines include DNA-based, viral vector-based, and recombinant protein-based vaccines [172,173,174]. Another innovative approach consists of the development of recombinant double-RBD (receptor binding domains) fusion molecules as potent bivalent subunit vaccines against the biotoxin BoNT and the tetanus toxin TeNT. The recombinant THc (TeNT heavy chain)-linker-AHc (BoNT/A heavy chain) and AHc-linker-THc molecules were strong and effective bivalent vaccines, protecting against two biotoxins simultaneously in vivo [175].

## 5. Plague

### 5.1. Background

*Yersinia pestis* is the agent of plague, an infectious disease responsible for more than 200 million deaths worldwide during the three major pandemics in history. Its reservoir is small rodents, with humans being accidental hosts who are often infected through flea bites. Today, the bacterium is found particularly in rural areas of Africa, Asia and the United States. Recent outbreaks, such as those in Madagascar in 2017, make plague a major public health problem [176].

*Yersinia pestis* can cause three forms of disease: bubonic, septicemic and pneumonic plague. Other forms can occur but remain rare (pharyngeal and meningeal). The most common clinical form in humans is bubonic plague, which is transmitted to the individual via an infected flea bite or by direct contamination of a skin lesion. The bacterium is phagocytized by macrophages and neutrophil cells and spreads to the nearest lymphoid nodule where it multiplies. The first symptoms appear 2 to 8 days after infection. Infected nodules, called buboes, can become necrotic if left untreated, causing hemorrhage and death in 50–60% of cases [177,178]. With treatment, mortality decreases to 13% [179]. Septicemic forms are caused when the bacteria pass into the bloodstream and begin multiplying subsequent to the bubonic form of the disease or when the bacteria are introduced directly into the bloodstream [178,180]. In the absence of treatment, the mortality rate is 100% and remains high (40%) even with antibiotic therapy. This failure rate is notably related to the late diagnosis of the disease for the primary forms of sepsis [175].

The primary pneumonic form is developed following inhalation of the bacteria transmitted from an animal or a person. The first symptoms appear in 1 to 3 days, and without prompt treatment, the disease is fatal [177,181]. This is the only form that can be transmitted from person to person, making it the most dangerous, although cases remain rare [177,182].

Recent advances in ancient genome sequencing and phylogenetic studies have established that *Yersinia pestis* diverged from *Yersinia pseudotuberculosis* only 5000 years ago [183,184,185].

Sequence comparison of the strains *Y. pestis* CO92 and *Y. pseudotuberculosis* IP32953 revealed that 75% of the chromosomal genes have a 97% or higher nucleotide identity [186]. This study also showed that *Y. pestis* lost the function of 525 genes (317 sequence losses and 208 became pseudogenes) and acquired 32 [187]. The 32 acquired genes in Yersinia pestis are grouped into six clusters with diverse characteristics. Four clusters encode largely hypothetical proteins with limited similarity to known proteins, except for a methylase. One cluster contains bacteriophage-related genes, suggesting horizontal gene transfer, while another encodes putative membrane proteins, a translation initiation inhibitor, and conserved hypothetical proteins. No clear virulence factors were identified, but their roles in pathogenicity and adaptation, such as in host interaction or immune evasion, warrant further investigation.

*Y. pestis* has three plasmids, one of which is common to the other two pathogenic species. This plasmid, pCD1 (also known as pCaD and pYV), is 68–75 kb in size and contains a low-calcium response system (LCRS), regulatory genes that control the expression of virulence proteins and a Type III secretion system. The smallest plasmid, 9.5 kb in size, is specific to *Y. pestis* and is called pPCP1 (also known as pPla and pPst). It carries genes encoding a plasminogen activator (pla), a bacteriocin called pesticin (pst) and a pesticin immunity protein (pim). The third plasmid pMT1 (also known as pFra), which is 60–280 kb in size, is sometimes integrated into the genome. It carries a gene encoding a phospholipase D (ymt) and a gene encoding the F1 fraction, a polypeptide constituting the capsule (caf1) [188]. These last two plasmids are present at 150–200 and one–two copies per bacterium respectively [189].

The identification of *Y. pestis* by classical bacteriological methods, combining culture and microscopy, remains the gold standard to make a diagnosis of plague. It is accompanied by an antibiogram to determine the sensitivity to the antibiotics proposed in therapeutic strategies. However, these techniques are rarely available in remote rural areas where the bacterium persists in an endemic state and a screening strategy using rapid diagnostic methods is needed [190]. The F1 antigen-based rapid diagnostic test (RDT) is validated for the diagnosis of bubonic plague and has been used on sputum for the diagnosis of pneumonic plague [191]. Serological testing, like ELISA, although impractical in the field, nevertheless remains an effective technique for retrospective confirmation. However, with these diagnostic tests, false-negative results can occur if the sample is collected at an early stage of infection. Confirmation should be sought using molecular biology methods, if possible, to improve the sensitivity of the diagnosis [189,192]. Other detection methods, such as mass spectrometry [193] and phage lysis assays, have been reported [194,195].

Because of its considerable capacity to spread, in case of exposure, preventive measures must be taken quickly. These consist, firstly, of informing people of the presence of zoonotic plague in their environment and advising them to take precautions against flea bites and not to handle animal carcasses.

In the event of an outbreak, the first step is to identify the most likely source of infection in the area where the human case(s) has been exposed, usually looking for clustered areas with a large number of small animal deaths. One of the first preventive measures is the elimination of fleas and then rodents; however, the inappropriate use of insecticides has already led to the development of resistance in fleas [196].

Persons in contact with infected cases who have been exposed to potentially infectious respiratory droplets are treated with oral Ciprofloxacin, Levofloxacin, Doxycycline, or Moxifloxacin for seven days or at least as long as they are exposed to infected patients according to CLSI recommendations [182].

Chemoprophylaxis should also be given to household members of patients with bubonic plague.

### 5.2. Therapy

#### 5.2.1. Antibodies

Currently, there are no antibodies approved for the treatment of plague. Only a limited number of recombinant and monoclonal antibodies have been developed. Immunization with F1 and V proteins has induced the production of neutralizing antibodies. The F1 and V antigens are only produced by *Y. pestis* when the bacteria enter the host and expression is triggered at 37 °C. F1 is a target for several neutralizing antibodies under development.

Liu et al. identified and developed an anti-F1 humanized murine monoclonal antibody F2H5, that provides complete protection against *Y. pestis* (28486528) [197]. 100 µg of F2H5 was administered (intravenous) 24 h before infection with 600 CFU of *Y. pestis* strain 141 (s.c. route). All mice treated survived. Based on a computational approach, they determined that F1 amino acids G104, E105 and N106 were critical for the binding of F2H5.

Xiao et al. isolated m252, a human anti-F1 protective antibody, from a naïve human FAB library [198]. m252 was administered one day before mice were challenged with *Y. pestis* strain CO92. The mice that received the antibody had a mean time to death of thirteen days (one out of six mice survived), whereas the other mice had a mean time to death of seven days. When mice received a second dose of antibodies five days after infection, the mean time to death was improved (20 days) and the number of mice that survived increased (five out of six).

Andrianaivoarimanana et al. also developed two monoclonal antibodies targeting the V antigen [199]. Mice were infected with 95 CFU of *Y. pestis* strain 10–21/S. mAb7.3 and mAb 29.3 were administered 24 h before or after infection. Fourteen days after infection, all mice administered mAb 7.3 and four out of five mice administered mAb 29.3 survived whether the antibody was administered before or after challenge. All mice that did not receive antibodies died.

#### 5.2.2. Antibiotherapy

Although plague is rapidly fatal, especially in the pulmonary form, it is easily controlled with antibiotic treatment including aminoglycosides, cyclines, fluoroquinolones and sulfonamides. Prompt and appropriate antibiotic therapy is still necessary to improve patient condition and reduce the mortality rate. Indeed, a study of 762 published clinical cases showed that mortality rates were higher when people were treated three or more days after the onset of symptoms [182].

The choice of antibiotic treatment depends on the clinical syndrome but also on the possibility of a bioterrorist attack. The CDC published guidelines on the treatment of plague in 2021 recommending the use of Ciprofloxacin, Levofloxacin, Moxifloxacin, Doxycycline (for bubonic and pharyngeal forms), Gentamicin and Streptomycin depending on the patient’s regimen and status (adult, child, pregnancy) [182]. Tetracyclines, Chloramphenicol and Trimethoprim Sulfamethoxazole should be considered as options due to their side effects [200].

In the management of septic patients, sometimes in severe states and at risk of multivisceral failure and shock, supportive care should not be neglected. In addition, certain old or innovative therapies such as immunotherapy (serotherapy, antibody therapy), phage therapy and the use of bacteriocin or virulence inhibitors, could reinforce the therapeutic arsenal of the future [201].

Antibiotic therapy is an effective treatment for *Yersinia pestis*. However, forms of resistance have been observed, mainly from the Madagascar region.

Five strains showing resistance to antibiotics have been isolated there, four of them human and one from a rat [202]. One of them (strain 17/95), isolated in 1995 from a human, has a highly transferable plasmid (plasmid IP1202), conferring resistance to Ampicillin, Chloramphenicol, Kanamycin, Streptomycin, Spectinomycin, sulfonamides, Tetracycline and Minocycline [203]. Also in 1995, a second strain called 16/95, carrying a highly self-conjugating plasmid (plasmid IP1203) containing a Streptomycin-resistant gene was isolated [204]. A plasmid (plasmid IP2180H) was also responsible for the Doxycycline resistance of the strain isolated from a rat in Antananarivo in 1998 [205].

The other two strains identified in Madagascar, 12/87 and 56/13–59/13, show Streptomycin resistance linked to mutations in the rpsI gene [202]. The same resistance mechanism is found in strain S19960127 in China [202,206].

The origin of the multidrug resistance (Gentamicin, Tetracycline, Doxycycline, Trimethoprim-Sulfamethoxazole and Chloramphenicol) in the strain collected in 2000 from a wild rodent in Mongolia has not been clarified, and the underlying mechanisms have not been characterized [207].

This, together with the global epidemic of antibiotic resistance, raises concerns about the emergence and possible spread of multidrug-resistant strains of *Y. pestis*.

#### 5.2.3. Chemical Inhibitors

For years, research teams have been attempting to develop agents targeting enzymes involved in the biosynthesis of Gram-negative bacterial membranes to combat infections. This effort has resulted in the engineering of molecules that inhibit uridine diphosphate-3-O-(R-3-hydroxymyristoyl)-N-acetyl-D-glucosamine deacetylase (LpxC), the enzyme that catalyzes the first step in lipid A biosynthesis, effective against a range of Gram-negative clinical isolates, including multi-resistant strains. Some of these inhibitors also show promise against *Y. pestis*. For instance, LPC-069 cured bubonic plague in mice but required a drastic regimen. While LpxC inhibitors hold potential for plague treatment, more potent molecules are needed, especially against multi-drug resistant strains. Their efficacy in treating pneumonic plague remains untested [208].

Synthetic cationic peptides such as LL-37 offer potential for the treatment of plague. [209]. In addition, the combination of broad-spectrum antimicrobial peptides with antibiotics such as tetracycline, minocycline or tigecycline has shown promise against *Y. pestis* in vitro. However, their efficacy in the treatment of plague, alone or in combination with other drugs, has yet to be validated in appropriate animal models [208].

#### 5.2.4. Vaccine

There are currently no vaccines approved for the treatment of plague. Several vaccines were developed, but their developments were stopped because they induced serious adverse effects. Other vaccines were also developed, but they were not effective against pulmonary plague, that is, the deadliest form of plague. Vaccine development for *Y*. *pestis* has mainly focused on the fraction 1 capsular antigen (F1). The low-calcium response V antigen (LcrV) and other antigens have been investigated as vaccine targets, but the results were less promising. This is partly because a specific portion of the protein adversely affects the host’s immune response [210]. F1 and LcrV were studied alone, as a co-mixture or as a fusion protein. Several studies determined that combinations of proteins induced better protection in animal models [211,212].

Vaccination with subunit vaccine candidates is considered to be an effective strategy for long-term protection. rF1V is a recombinant vaccine, based on the fusion of the F1 and LcrV proteins. The antigen is composed of the N-terminus of the V protein, fused to the C-terminus of the F1 protein [213]. Adjuvanted with Alhydrogel, this vaccine induced similar protection against pneumonic plague in mouse and macaque models to that of the mixture of both proteins. It was safe in an animal model and clinical phase 1 study, and it induced a robust antibody response, but a poor cell-mediated immune response [214]. To improve the immunogenicity of the vaccine, Alhydrogel was replaced by the adjuvant CpG 1018^®^, a TLR9 agonist promoting T-helper 1 immune responses. rf1V-1018 is currently in a clinical phase 2 study. Two intramuscular doses of this vaccine (one month apart) induced protection, whereas, without CpG 1018^®^, three doses were required over six months [215].

Moore et al. developed a dual-route vaccination approach [216]. They developed a vaccine (VypVaxDuo) that is composed of the free association of the F1 and V proteins. VypVaxDuo has been designed for administration via a subcutaneous priming dose followed by a single oral booster dose and has been demonstrated to induce early onset immunity 14 days after the primary immunization. Mice were immunized in the dual route dosing regimen on D0. 21 mice were challenged with 2.2 × 10^4^ MLD or 2.2 × 10^6^ MLD of *Y. pestis* CO92 (s.c. route) on day 46. Whereas all unvaccinated mice died, respectively, 100% and 90% of vaccinated mice survived.

Attenuated vaccines are another approach studied. NlpD is a lipoprotein that is an essential virulence factor of *Y. pestis* for the development of bubonic and pneumonic plague; Dentovskaya et al. generated several ΔnlpD mutants [217]. Immunization of mice with these mutants induces immunity 105 times more potent than that induced by the administration of the EV line NIIEG vaccinal strain of reference. Mice and guinea pigs were challenged 21 days after vaccination. The animals were challenged (s.c. route) with 200 LD_100_ of a virulent strain of *Y. pestis*, strain 231 (1 DCL was equal to 10 CFU in mice and 100 CFU in guinea pigs). Vaccination induced protection in the mice model, but no significant protection was observed in the guinea pig model. Without additional modifications that would increase their immunogenicity in guinea pigs, Δ*nlpD* mutants are not promising candidates for live plague vaccines.

Rapid post-exposition vaccination with the live-attenuated EV76 vaccine strain is another strategy. C57BL/6 mice were immunized (s.c. route) with 100 LD_50_ of the virulent strain KIM53. 1  ×  10^7^ CFU of EV76 was administered simultaneously or after 5 h. Protection of 91% and 34%, respectively, was observed when challenged intranasally (pneumonic plague model). Protection in the pneumonic plague model was also analyzed. Mice were immunized (s.c.) with 1  ×  10^7^ CFU of the EV76 vaccinal strain and an intranasal challenge was realized with 1  ×  10^4^ CFU (10 LD_50_) of the lethal strain KIM53. When the challenge was realized 2 days post EV76 administration, 60% of the mice survived. When the challenge was realized at the same time, only an increase in time to death (3 to 6.8 days) was observed.

Kon et al. developed a single-dose F1-based mRNA-LNP (lipid nanoparticle) vaccine [218]. SP-cp-caf1 mRNA was encapsulated in LNP. In vitro, the formulation induced the expression of the F1 protein. Mice were immunized once with 5 µg of the mRNA vaccine. Vaccinated animals were inoculated subcutaneously with 100 LD_50_ of the fully virulent *Y. pestis* strain Kim53. All vaccinated mice survived.

Recently, Demeure et al. have created an oral vaccine for plague using a genetically attenuated strain of *Yersinia pseudotuberculosis* producing the *Yersinia pestis* F1 antigen. This vaccine has shown promising outcomes in rodent studies, paving the way for potential testing in non-human primates [219]. Similarly, Majumder and Olson developed a new attenuated Yersinia pseudotuberculosis strain, Yptb1, engineered to express Yersinia pestis antigens. They demonstrated that an oral prime-boost immunization with Yptb1(pYA5199) stimulates a strong immune response and provides complete short-term and long-term protection against pneumonic plague in mice and rats [210].

## 6. Tularemia

### 6.1. Background

Tularemia is a zoonotic disease caused by the bacterium *Francisella tularensis*, a small, facultative, intracellular Gram-negative coccobacillus that is highly infectious to humans. It is considered one of the most infectious bacteria, with the ability to cause human infection with only 10 organisms. Its mortality rate is estimated between 30 and 60% without treatment [220]. Two subspecies are responsible for tularemia: *Francisella tularensis* subsp. *holarctica* (type B) located in the Northern Hemisphere and Australia, and *F. tularensis* subsp. *tularensis* (type A), the most virulent, mainly found in North America. Its main natural reservoir is made up of numerous animal species (lagomorphs (rabbits, hares), small wild rodents, but also sometimes cats and dogs). There is also a reservoir in arthropods (Ixodidae ticks) and an environmental reservoir where the bacteria can survive for extended periods in hydro-telluric habitats (WHO 2007). Humans become infected through contact with the animal reservoir (via skin inoculation, ingestion or inhalation of aerosols), through arthropod vectors (tick bites, and less commonly, bites from mosquitoes or horseflies in some areas), or from environmental sources (water, moist soil), where the bacteria can persist for several months (WHO 2007).

Human-to-human transmission is insignificant [221]. Incubation time varies from three to five days but can extend to 21 days (WHO 2007). The disease begins with the onset of flu-like symptoms, such as decreased general condition, fever, chills, headache, myalgia and anorexia. In addition, other manifestations depend on the mode of inoculation of the disease. Six clinical forms have been identified, depending on the mode of transmission and the point of entry of the bacteria (WHO 2007) [222]. The ulcero-glandular form is the predominant form in Europe, with 90% of cases caused by *Francisella tularensis holartica*. This form is contracted by direct contact with the infected animal or by vector transmission after the bacterium has entered the body through a skin lesion, which may develop into an ulcer. The ulcer may remain discreet and heal within a week and may be mistaken for a tick or mosquito bite. The lesion may become necrotic, and adenopathy may develop in the lymphatic drainage areas of the wound. Ulcerations are mainly located on the lower and upper limbs (Evan, 1985). Without antibiotic treatment after a 5- to 10-day infection, the lesion may necrose and adenopathy may develop in the lymphatic drainage areas of the wound, resulting in suppuration in 30 to 40 percent of cases, the most severe complication of tularemia caused by *Francisella tularensis holartica* [223,224].

The glandular form manifests similarly to the ulcero-glandular form, although without an identified cutaneous entry point.

In 1–4% of cases, direct contamination of the eyes with the bacteria results in oculoglandular tularemia, which is caused by splashing infected material into the eye or by contact with contaminated fingers, and may result in conjunctivitis, eyelid swelling, photophobia and purulent secretions [225].

The oropharyngeal form caused by the ingestion of contaminated water or food, causing pharyngitis, stomatitis and cervical lymphadenopathy, occurs in 5% of cases.

The typhoid form, although rare, is one of the most severe forms of tularemia, causing severe systemic symptoms with a high mortality rate in 50% of cases [226].

Pulmonary tularemia is the most severe form of tularemia, with type A strains causing up to 30% lethality if left untreated [227]. Symptoms include coughing, chest pain and difficulty breathing. This form results from the inhalation of dust or aerosols containing the bacteria. It can also occur when other forms of tularemia (e.g., ulcerative) are not treated and the bacteria spreads through the bloodstream to the lungs.

In the *Francisellaceae* family, the single genus *Francisella* contains around ten species, including *F. tularensis*, *F. philomiragia*, *F. novicida* and *F. noatunensis*, which differ according to their reservoir, metabolism and virulence.

*Francisella tularensis* is the major human pathogenic species of the *Francisella* genus. Its genome is 1.9 Mb in size, and is contained in a single, stable circular chromosome. It is highly conserved and rich in AT.

The three subspecies of *F. tularensis*, *F. tularensis* subsp. *Tularensis*, *F. tularensis* subsp. *Holarctica* and *F. tularensis* subsp. *Mediasiatica*, although differing markedly in virulence and originating from different regions of the world, are very closely phylogenetically related and are antigenically similar. Only the subspecies *Holartica* and *Tularensis* are responsible for tularemia. Depending on the literature, *Francisella novicida* appears as a species in its own right or as a subspecies of *Francisella tularensis* [228].

Despite the highly conserved genome of the species (>99.2% identity between subspecies), recent whole genome sequencing techniques, coupled with canSNP and INDEL identification analyses, have allowed *F. tularensis* strains to be divided into clades and subclades [229].

In *Francisella tularensis* subsp. *Tularensis* SCHUS4, 1281 identified genes have homologs in other bacteria of the same subclass. Genes encoding type IV pili, a surface polysaccharide and iron acquisition systems were identified. Several genes associated with virulence and the escape of the bacterium from the phagosome into the host cell have been localized in a pathogenicity island that is duplicated in the genome [230].

Populations at particular risk include all persons exposed to small wild mammal droppings, tick bites, and game: hunters, forest workers, hikers, and rural residents, especially in endemic areas.

Measures to prevent the disease include wearing long-sleeved, long-legged clothing for recreational or occupational activities in the forest and checking for ticks on the skin after returning from outdoor activities. In addition, it is recommended to avoid handling dead animals. Wild game meat should be cooked very thoroughly [231].

For laboratory personnel, in case of accidental exposure, treatment with Ciprofloxacin or Doxycycline is initiated. If exposure is unlikely, increased vigilance may be sufficient. In the event of accidental aerosolized spread of the bacteria, personnel should be alert for the development of fever within 14 days of exposure, and treatment should be initiated if necessary (WHO 2007).

Tularemia can be difficult to diagnose. It is a rare disease, and symptoms may be confused with other more common diseases, resulting in a delay in identifying the pathogen. For this reason, it is important to consider the patient’s environment (likely exposure from arthropod bites or contact with sick or dead animals).

Diagnosis of tularemia is most commonly performed by regulatory techniques, such as the microagglutination technique and indirect immunofluorescence [220]. ELISA and western blot techniques, often utilizing antigens from the less virulent LVS strain of *Francisella tularensis Holartica*, are also used. The ELISA technique may face challenges due to the lack of standardization and potential cross-reactivity with other organisms like Brucella species.

The tube agglutination test, known for its affordability and simplicity, is commonly employed. The latex agglutination test, a recent development, offers specificity, sensitivity, rapidity and ease of use, making it suitable for routine tularemia diagnosis, especially in mobile laboratories [221].

A recommended two-step approach involves initial ELISA screening tests followed by confirmatory immunoblot tests for serological diagnosis [222]. The isolation of the bacterium through culture, although challenging and limited to early disease stages, remains the gold standard for identification, requiring a high-containment level 3 laboratory due to F. tularensis’ high pathogenicity (WHO 2007). Identification can be supplemented with MALDI-TOF mass spectrometry, though it does not allow for subspecies identification [223].

Molecular methods, such as PCR, play a crucial role in research by directly identifying F. tularensis from human, animal and environmental samples through target nucleic acid sequence amplification with specific primers. Due to genomic similarities among Francisella species, careful primer selection is essential for specificity [224]. Combining multiple PCR tests is recommended to enhance sensitivity and specificity.

The GenXpert cartridge-based test represents a new advancement, offering the rapid detection of F. tularensis directly from whole blood during early infection stages, surpassing other methods in sensitivity [1,225,228,229,230,231,232].

The World Health Organization (WHO) has outlined two criteria for defining suspected tularemia cases (WHO 2007). They are associated with clinical manifestations consistent with tularemia, either coupled with a history of exposure to risk factors (e.g., contact with vector animal) or with a positive tularemia test (e.g., elevated single-serum antibody titer or detection of antigen). The confirmed case definition is the isolation and identification of *F. tularensis*, or a rise in serum antibody titer to *F. tularensis* of at least a fourfold factor between the acute and convalescent phases.

*Francisella tularensis* is considered a potential biowarfare agent due to its extreme infectivity (requiring only 10 organisms to cause disease), ease of spread and significant capacity to cause morbidity and mortality. During World War II, the feasibility of using it as a biological weapon was explored by both Japan and the United States and its allies. Tularemia was among the biowarfare agents accumulated by the US military in the late 1960s, prior to its destruction in 1973. The Soviet Union continued to develop antibiotic- and vaccine-resistant strains into the early 1990s [232], although its use as a biological weapon dates back to the 14th century, when tularemia epidemics were deliberately spread in Anatolia [1]. It is believed that, among the various methods of weaponizing *F. tularensis*, aerosol dispersal would have the most severe medical and public health impacts. A World Health Organization (WHO) expert committee reported in 1970 that aerosolizing 50 kg of virulent *F. tularensis* over a metropolitan area with a population of 5 million could result in approximately 250,000 incapacitated individuals, including 19,000 fatalities [226].

### 6.2. Therapy

#### 6.2.1. Antibodies

No antibodies have been approved for the treatment of Tularemia, and only limited data presenting the development of therapeutic antibodies are available.

The use of a FcyR-targeted monoclonal antibody (mAb)-iFt gives only partial protection against a *F. tularensis Tularensis* SCHUS4 challenge in mice with an enhanced humoral and cellular immune response [233].

Monoclonal antibodies (mAbs) targeting components of the LVS strain have demonstrated the ability to protect mice that are infected with this strain. However, their effectiveness was notably reduced when mice were infected with the Schu S4 strain of F. tularensis type A [234]. In another study, Klimpel et al. demonstrated that mice infected intranasally with the Schu S4 strain and later treated with levofloxacin developed protective immunity [235].

Regarding subunit vaccine development, research is focused on bacterial surface proteins and LPS from the bacterial outer membrane. Thus, despite in vitro results showing a boosted immune response via LPS injections in animals or humans, the protection of the whole organism is not sufficiently effective compared to LVS injections [236].

An immunogenic LPS derived from *F. tularensis Tularensis* SCHUS4 does not induce protection against a challenge with the F. tularensis Tularensis SCHUS4 strain, whereas protection is observed against a challenge with *Francisella tularensis Holartica* [237,238]. The vaccination of mice with the polysaccharide capsule O antigen improves protection against the LVS, but not against more virulent strains [239,240].

#### 6.2.2. Antibiotherapy

Tularemia is a disease that must be treated with antibiotics. The early recognition and appropriate treatment of the disease are essential.

Aminoglycosides (Streptomycin and Gentamicin), fluoroquinolones (Ciprofloxacin) and tetracyclines (Doxycycline) are the three classes of antibiotics recommended by the WHO (WHO 2007). The choice of treatment depends on the patient’s condition and the clinical form of the infection. Regular serology is recommended. Treatment usually lasts 10 to 21 days, depending on the stage of the disease and the antibiotics used. The Civilian Biodefense Task Force has defined antibiotic therapy according to two situations, which are firstly contained casualty cases, where Streptomycin and Gentamycin are used preferentially as first-line parenteral antibiotics, with Doxycycline and Ciprofloxacin as alternatives. Failures and relapses are more frequent with bacteriostatic antibiotics, tetracyclines and Chloramphenicol. Secondly, for mass casualties, as well as the most severe cases, oral Doxycycline and Ciprofloxacin are recommended [226].

Antibiotic susceptibility testing is not recommended in cases of infection because of the lack of natural resistance forms of *Francisella tularensis* to the antibiotics of choice, and facilities may not have adequate biosafety conditions. However, because of the potential use of *F. tularensis* as a bioweapon, the emergence of resistance must be monitored in reference laboratories. Various studies have indeed shown that forms of *Francisella tularensis* that are resistant to Ciprofloxacin and Streptomycin are possible [241,242,243].

The natural resistance of *Francisella tularensis* to β-lactamine is due to the production of β-lactamase and the poor penetration of the antibiotic into cells [244,245].

*Francisella tularensis Holartica* biovar 2 is resistant to Erythromycin due to the presence of a mutation in the gene encoding 23S ribosomal RNA [246,247].

#### 6.2.3. Chemical Inhibitors

Cationic antimicrobial peptides (eCAPs) exhibit activity against various bacterial pathogens, including those resistant to multiple drugs. Researchers compared two key synthetic eCAPs, WLBU2 and WR12, to the human cathelicidin peptide (LL-37) in their effectiveness against several highly virulent bacteria, such as Francisella tularensis and Yersinia pestis.

Both WLBU2 and WR12 demonstrated superior bactericidal activity to LL-37 against these two strains. These findings illustrate the promising therapeutic capacity of these peptides as effective antimicrobials with broad-spectrum activity against highly virulent bacterial infection [209].

The pathogenicity of *F. tularensis* is closely linked to its ability to evade or suppress the host immune response while growing in phagocytic cells. Several studies suggest that increasing levels of pro-inflammatory cytokines during the early stages of infection may be beneficial by activating the innate immune system at an early stage. By doing so, it may be possible to improve treatment outcomes, thereby reducing antibiotic doses and treatment times. Lembo et al. have developed a synthetic TLR4 agonist, aminoalkyl glucosaminide phosphate (AGP), that reduces the replication of the bacterium in the lung, liver and spleen and improves survival in animals infected with F. novicida [248]. Pyles et al. have synthesized a TLR3-activating synthetic double-stranded RNA ligand called polyinosine-polycytosine (poly(I:C)) that induces an early innate immune response in mice infected with *F. tularensis* LVS and SchuS4 strains [249].

#### 6.2.4. Vaccines

Tularemia vaccine development began in the 1940s in Western countries when *Francisella tularensis* was considered a potential bioweapon.

Currently, the only vaccine available to combat tularemia is the attenuated live vaccine strain (LVS). In the United States, it is available to at-risk personnel through the Special Immunization Program administered by the Department of Defense [250].

This vaccine strain was isolated by the US Army from a strain of attenuated virulence obtained by the USSR prior to the 1950s by the successive passages of a *Francisella tularensis Holartica* strain in rich media [251]. This strain was administered on a large scale to the Russian population as early as 1946, and also in US government research institutes in the 1960s and 1970s with scarification [252]. This strain as a vaccine, however, has its limitations, and is therefore not approved by the FDA [253]. Indeed, the mechanisms of virulence attenuation and infection remain unknown, and the reversion of virulence may be possible. The cultures of the LVS are also unstable [251,254]. In addition, the LVS cannot completely prevent pneumonic infection from the more virulent type A bacteria. This limits its scope of action.

Despite this, the LVS remains the most studied strain, and many trials are being conducted with the LVS, *F. tularensis novicida* and *F. tularensis Tularensis* SCHUS4 strains deleted in one or more genes involved in virulence to obtain a safer and more stable vaccine [253]. A new production of the LVS (DC-LVS) by the US Army under good manufacturing conditions is in phase 2 clinical evaluation [255].

The development of a safe vaccine that induces rapid and effective protection is essential but remains a challenge. Many trials are effective against type B strains but are insufficient for the more virulent type A strain.

Thus, vaccination with the killed vaccine does not prevent local lesions after a skin challenge but does reduce systemic manifestations of infection, and it does not confer any protection after a respiratory challenge [255]. The use of adjuvants such as Freund’s adjuvant does not increase the efficacy of the inactivated strain [256,257]. Although IL-12 improves the clearance of the inactivated LVS, no protection is achieved on a SCHUS4 challenge [257,258].

Research on the development of a new, safer and more effective vaccine must continue by ensuring not only antibody generation but also the induction of a T-cell response [253,259,260].

## 7. Viral Hemorrhagic Fevers

### 7.1. Background

Viral hemorrhagic fevers (VHFs) are a group of viruses inducing febrile illnesses. VHFs are caused by four RNA virus families: *Filoviridae*, *Arenaviridae*, *Flaviviridae* and *Bunyaviridae*. The most well-known and feared viruses are the Ebola and Marburg viruses of the *Filoviridae* family, as well as the Lassa and Machupo viruses of the *Arenaviridae* family. Most VHF viruses are characterized by outbreaks which occur irregularly and are hard to anticipate. VHFs are mainly zoonotic and/or vectorized by arthropods, but when a person is infected by animals/arthropods, the virus can spread rapidly and easily among human populations, mainly through direct contact with body fluids such as blood, or via fomite or aerosol. VHFs can be endemic and epidemic in some areas of the world, such as Ebola in Guinea and CCHF in Africa, the Middle East, the Balkans and Asia. South Africa is the only country where four different VHF outbreaks have been reported (CCHF, RVF, LHF and MVD) [62]. Despite rapid diagnosis and treatment, mortality rates can be greater than 90% (Table 2). Mortality is highly dependent on the time between contamination and the administration of appropriate treatment in an intensive care unit. Many VHF outbreaks occur in poor countries, which can also explain the mortality rates. The rapid spread of the virus may also be due to funeral rites in some countries that require proximity to the bodies of the dead. To prevent the spread of the virus, hospitalization in a BSL3 area and the rigorous application of biosecurity procedures are required. Before 2014, all humanitarian staff contaminated with Ebola were treated in Africa, despite limited medical facilities. In 2014, for the first time, an American doctor contaminated in Sierra Leone was repatriated to the USA. Spain and France have also repatriated medical staff to their respective countries to provide better medical care, despite the risk of potential outbreaks outside of Africa. On 30 September 2014, the CDC confirmed the first travel-associated case of EVD diagnosed in the USA in a man coming back from West Africa. The patient died on 8 October 2014, and two healthcare workers who had cared for him in Dallas tested positive for EVD. There are almost no specific drugs, so treatment is mainly symptomatic. All VHF viruses are classified as level 4 pathogens, which limits the number of laboratories in the world that can develop new drugs. Here, we will focus on treatments against Ebola.

### 7.2. Therapy

There is almost no specific treatment for VHF. Recovery depends mainly on supportive clinical care and the treatment of complications. The major issue during hospitalization is the implementation of strict infection control measures with the use of appropriate personal protective equipment, to prevent the contamination of hospital staff and other patients. Hospitalization in a level 3 biosecurity zone (with directionally pressurized rooms) is essential, but such infrastructure is rare.

#### 7.2.1. Antibodies

In the absence of more reliable alternatives, for a long time, “specific” treatment of VHF was based on the intravenous administration of immune plasma or whole blood obtained from convalescent donors.

During the 2014 Ebola pandemic, Zmapp (Mapp Biopharmaceutical) was tested as an investigational new drug. Zmapp is composed of chimeric antibodies. No significant protection was observed, and this drug candidate was stopped in 2016 [267].

The EVD outbreak between 2013 and 2016 was an opportunity to evaluate specific antiviral drugs that were not tested in sick peoples until then. Unfortunately, the clinical trials evaluating Favipiravir and Zmapp started too late in the outbreak to yield robust results. Most of the patients evacuated from Africa to Europe or North America received one or more investigational therapies and two-thirds of them received at least two experimental drugs [268].

Two drugs are approved by the US Food and Drug Administration (FDA) to treat EVD caused by Ebola Zaire. These two treatments were compared in a randomized controlled trial (Pamoja Tulinde Maisha (PALM) study) [268].

REGN-EB3 (Inmazeb™, Regeneron Pharmaceuticals) was approved in October 2020. Inmazeb™ is an oligoclonal drug composed of three fully human monoclonal antibodies (IgG1): Atoltivimab (REGN3470), Odesivimab (REGN3471), and Maftivimab (REGN3479). All of them target the Ebola virus glycoprotein, are non-competitive and have affinities of 7.74, 8.42 and 2.97 nM, respectively, for the recombinant histidine-tagged Makona strain Ebola virus glycoprotein ectodomain protein. In vitro, the IC_50_ of Inmazeb is 0.39 nmol·L^−1^. In a rhesus macaque model of Ebola virus infections, three doses of 50 mg·kg^−1^ (1:1:1 equimolar ratio) or a single dose of 150 mg·kg^−1^ of Inmazeb^TM^ on days 5, 8 and 11 post-infection completely protect the animals from death, after a challenge with 1000 PFU of the Kikwit strain of EBOV by intramuscular injection [269,270].

Ansuvimab (Ansuvimab-zykl, mAb114, Ebanga™) is the most recent FDA-approved drug for treating Ebola. It was approved by the FDA in December 2020. This monoclonal antibody (IgG1) was originally isolated from the blood of a survivor of the 1995 outbreak of Ebola virus disease in Kikwit, Democratic Republic of the Congo. The recommended dosage of Ansuvimab is 50 mg·kg^−1^ administered intravenously [271]. In an animal study, macaques were challenged (intramuscularly) with 1000 PFU of Ebola virus. When mAB114 was administered intravenously (50 mg·kg^−1^ at 24, 48 and 72 h post-infection), all macaques survived the challenge. All macaques also survived when the treatment was started 120 h post-infection [272].

A meta-analysis determined that REGN-EB3 and mAb114 separately reduce mortality compared with ZMapp, Remdesivir or standard care in patients with Ebola virus disease. There is probably little to no difference between REGN-EB3 and mAb114 in the prevalence of serious adverse events. These findings suggest that REGN-EB3 and mAb114 should be used as a first-line treatment [273].

Ansuvimab and REGN-EB3 have shown acceptable tolerability profiles in patients enrolled in the phase II/III PALM trial, despite the assessment of adverse events or reactions that may have been confounded by symptoms of the Ebola virus infection. In particular, severe adverse events have been reported with REGN-EB3 and Ansuvimab, including death.

Both of these treatments, along with two others, were evaluated in a randomized controlled trial during the 2018–2020 Ebola outbreak in the Democratic Republic of the Congo. The overall survival rate was much higher for patients receiving either of the two treatments that are now approved by the FDA. Neither Inmazeb™ nor Ebanga™ have been evaluated for efficacy against species other than *Zaire ebolavirus*.

#### 7.2.2. Chemical Inhibitors

Ribavirin is currently the only antiviral that can be administered for the treatment of certain VHF diseases. Particularly, Ribavirin is the standard of care for treating Lassa fever. Ribavirin is a synthetic antiviral (a purine nucleoside analog) with antiviral activity against several families of both ribonucleic acid (RNA) and deoxyribonucleic acid (DNA) viruses, approved by the US Food and Drug Administration (FDA) for the treatment of chronic hepatitis C virus infection, in combination with other drugs. Ribavirin can be administered orally or intravenously. Nevertheless, for VHF treatment, Ribavirin is used as an emergency investigational new drug. Several systemic reviews have emphasized that the human clinical trial data supporting the use of Ribavirin for VHF treatments suffer from several serious flaws that render the results and conclusions unreliable [266,274]. Today, it is still not clear if Ribavirin provides a significant benefit in the treatment of Lassa fever. Some benefits may be observed, but they may be offset by the adverse effects. Systemic reviews have also pointed out that there is no strong proof of the efficacy of Ribavirin in the treatment of other VHFs, such as CCHF or hantavirus pulmonary syndrome.

Remdesivir (Gilead Sciences), a small-molecule nucleotide analog RNA polymerase inhibitor, has been administered to hundreds of patients under the monitored emergency use of unregistered and investigational interventions (MEURI) framework and in a randomized clinical trial [275]. Patients in the Remdesivir group received a loading dose on day 1 (200 mg in adults), followed by a daily maintenance dose (100 mg in adults) starting on day two continuing for nine to thirteen days, depending on viral load. Adverse effects were initially expected; Remdesivir failed to significantly cure patients, and significant toxicity was observed [276].

#### 7.2.3. Vaccine

Vaccines are approved only for the prevention of yellow fever, Argentine hemorrhagic fever and Ebola Zaire.

Vaccination against yellow fever with YF-Vax/Stamaril (17D-204 strain of yellow fever) is recommended for people with ages of 9 months and over who are traveling to or living in areas at risk for yellow fever virus transmission. A single shot is necessary, but a booster is recommended ten years after initial vaccination and for travelers to areas with ongoing outbreaks. The vaccine is safe and effective, but serious adverse events can occur following yellow fever vaccination. Therefore, people should only be vaccinated if they are at risk of exposure to yellow fever. Serious adverse effects include hypersensitivity reactions (1.3 cases per 100,000 doses administered), yellow fever vaccine-associated neurologic disease (2.2 per 100,000 doses) and yellow fever vaccine-associated viscerotropic disease (1.2 per 100,000 doses).

Argentine hemorrhagic fever is caused by the Junin virus. A live-attenuated vaccine, Candid#1, has been approved, but only in Argentina, particularly due to the potential for reversion at the attenuating locus, a phenylalanine-to-isoleucine substitution at position 427 in the GP2 subunit of the GPC envelope glycoprotein [277,278].

In 2019, the FDA approved the ERVEBO (Merck) Ebola vaccine for the prevention of Ebola Zaire in adults who will be exposed to Ebola virus (responders in outbreak areas, healthcare personnel at federally designated Ebola treatment centers in the United States and laboratory workers or other staff at Biosafety Level 4 facilities in the United States). In 2023, ERVEBO vaccination was approved for use in children aged 12 months and older in the USA.

## 8. Smallpox

### 8.1. Background

Smallpox is caused by the variola virus (VARV), a member of the *Orthopoxvirus* (OPV) genus of the *Poxviridae* family. Smallpox is one of the most devastating diseases known to humankind. Edward Jenner described smallpox as “the most dreadful scourge of the human species”. It was estimated that in the 20th century, more than 300 million people died from smallpox. It is suspected that this virus appeared between 1000 and 3000 years before the common era, as scars characteristic of the disease were observed in mummies dated to this period [279,280]. Nevertheless, the first documentation of smallpox dates back to the 4th century in China. By the mid-18th century, smallpox had been reported all over the world. The oldest available variola virus strains were isolated in the early 1940s [281]. Two distinct forms of smallpox have been observed: variola major with a case fatality rate of 20–30%, and variola minor (also referred to as Alastrim, Kaffir or Amass) with a case fatality rate of 1% or less. Among the six pathogens on the CDC’s A list, smallpox is atypical because it has been eradicated. The last natural case was reported in Somalia in 1977, and the last human case (lab-acquired) was reported in 1980. The eradication was officially declared by the World Health Organization in 1980. In 1984, systemic vaccination ceased, and it became limited to people at a high risk of exposure. To limit the risk of resurgence, the decision was made to destroy all stocks of variola virus. Only two stocks have been preserved to continue the development of medical countermeasures in case of resurgence: that of the State Research Center of Virology and Biotechnology Vector (Koltsovo, Russia), and that of the Centers for Disease Control and Prevention (Atlanta, GA, USA). The CDC classification is based on the potential consequences of natural or intentional resurgence among a naïve population.

### 8.2. Therapy

Because there are no longer smallpox cases and only two BSL4 (BioSafety Laboratory) in the world are authorized to work with the variola virus, the development of new drugs is limited. The approval of a new drug or of a new indication is generally done under the Animal Rule. The recent mpox pandemic has also demonstrated that there is an urgent need for the development of medical countermeasures against all *Orthopoxvirus* infections for humans.

#### 8.2.1. Antibodies

Vaccinia immune globulin intravenous (human) (referred to as VIGIV, VIG or CNJ-016^®^) is the only approved therapy for the treatment of the serious adverse effects resulting from vaccination. Serious adverse effects include eczema vaccinatum, progressive vaccinia, severe generalized vaccinia, vaccinia infections in individuals who have certain skin disorders and aberrant infections induced by the vaccinia virus. VIG is not approved for the treatment of smallpox and is not useful for the treatment of post-vaccinal encephalitis.

VIG is a passive immunotherapy, composed of the IgG fraction of pooled human plasma containing antibodies to the vaccinia virus, collected from healthy donors following their recent vaccination with the vaccinia virus vaccine [282]. Immunoglobulin represents 5% of the VIG solution. VIG should be administered at a dose of 6000 U·kg^−1^, as soon as symptoms appear, then an additional dose of 6000 U·kg^−1^ to 9000 U·kg^−1^ may be administered. Doses of up to 24,000 U·kg^−1^ appeared to be safe during the clinical trial (Expanded Access IND Protocol: Use of Vaccinia Immune Globulin Intravenous (VIGIV, CNJ-016) for Treatment of Human *Orthopoxvirus* Infection in Adults and Children).

VIGIV safety has not been studied in patients with smallpox or non-vaccinia *Orthopoxvirus* diseases. The existence of shared serological cross-reactivity and cross-neutralization between the *Orthopoxviruses*, has been observed. Thus, it is hypothesized that VIGIV may provide some amount of protection against smallpox or less virulent *Orthopoxvirus* such as the monkeypox virus. During the mpox pandemic, VIG was administered to some patients, in combination with Tecovirimat or additional antivirals, under EA-IND. The data obtained were unclear (Expanded Access IND Protocol: Use of Vaccinia Immune Globulin Intravenous (VIGIV, CNJ-016) for Treatment of Human Orthopoxvirus Infection in Adults and Children).

Several recombinant antibodies neutralizing smallpox have been developed, but none of them were developed clinically or approved for therapy. Gilchuk et al. developed a panel of human monoclonal antibodies that are cross-reactive with several *orthopoxviruses* [283]. The majority of purified mAbs reacted to one of the six VACV antigens that were previously reported as major targets for neutralizing Abs in mice or humans: A27, H3, D8, L1, B5 and A33, and the majority of them bind several *Orthopoxviruses*. (21198662 = moss 2011) Most of the antibodies neutralized the vaccinia virus, cowpox virus and monkeypox virus with IC_50_ values of individual mAbs, ranging from ~0.02 to 100 μg·mL^−1^. Higher neutralization was achieved with oligoclonal antibodies, and the mixture also neutralized the variola virus (mature virions).

#### 8.2.2. Chemical Inhibitors

The administration of antivirals is recommended if treatment with VIGIV (see below) alone is inadequate or if VIGIV is not readily available, three antivirals may be considered. Two molecules were recently approved for the treatment of smallpox, and one is under expanded access to investigational new drug (IND) protocols.

Tecovirimat (TPOXX, ST-246) is the first antiviral approved by the FDA (2018) for the treatment of adult and pediatric smallpox infection and could be used under an expanded access investigational new drug protocol for the treatment of adverse reactions secondary to continued vaccinia virus replication after smallpox vaccination. This drug targets the virus VP37 (homolog of the F13 protein) phospholipase and acts as an inhibitor of virus egress and blocks the formation of enveloped forms of *Orthopoxviruses*. Its efficacy against smallpox was investigated in several animal models, particularly in non-human primates [284,285]. Eight monkeys were infected (IV route) with 1 × 10^8^ PFU of the Harper strain of VAR, then 300 mg·kg^−1^ of Tecovirimat was administered orally immediately or 24 h after infection. All non-human primates that received Tecovirimat survived. In another study, no interactions with the Dryvax or ACAM200 vaccines were observed, so Tecovirimat can be administered simultaneously.

During the mpox outbreak, Tecovirimat was administered in the United States on a large scale for the first time. The MPXV F13L gene homolog encodes the target of Tecovirimat, and single amino acid changes in F13 are known to cause resistance to Tecovirimat. Smith et al. identified 11 mutations previously reported to cause resistance and 13 novel mutations. A total of 124 isolates were analyzed from 68 patients, and 96 isolates from 46 patients were found to have resistant phenotypes [286]. The development of several medical countermeasures is thus essential for overcoming potential resistance.

Cidofovir is a nucleoside analog that selectively inhibits the viral DNA polymerase and reduces the replication of VARV in vitro [280]. It was initially approved for the treatment of HCMV retinitis in HIV patients. When administered before the onset of the smallpox rash, Cidofovir could prevent mortality, despite some serious adverse effects such as nephrotoxicity and it could induce cancer in animals [287]. However, there are only limited data on its effectiveness in the treatment of vaccinia-related complications in humans and in the treatment of smallpox disease. Cidofovir can only be used under an expanded access IND protocol for the treatment of complications that might arise from vaccinia virus vaccination.

Brincidofovir (CMX001, TEMBEXA) is a lipid analog of Cidofovir active against double-stranded DNA viruses, including *orthopoxviruses*. It can be administered orally, and no nephrotoxicity has been reported [288,289]. Brincidofovir was effective in a rabbit model infected (intradermal) with rabbitpox and in mice infected with ectromelia (intranasal). In 2019 and 2022, systematic reviews reported five human cases of *orthopoxvirus* infections treated with Brincidofovir. No relevant conclusions about the efficacy of Brincidofovir against smallpox in humans can be drawn from these reviews, but Brincidofovir was approved by the FDA in 2021 for the treatment of smallpox infections, based on animal studies [290,291]. The administration of Brincidofovir is generally recommended as a second-line treatment or in combination with Tecovirimat.

For all three molecules, the only available data concerning their protection against the monkeypox virus is limited, but several studies are in progress.

#### 8.2.3. Vaccines

Humans are the only reservoir of smallpox, so a global vaccination program was conceived of to completely eradicate the virus. Throughout this program, three generations of vaccines were developed using different VACV strains. Smallpox vaccination is possible pre-and post-exposure. Post-exposure vaccination should ideally be administered within 4 days of exposure to prevent infection, but it can be used up to 14 days after exposure to decrease the severity of the disease. Post-exposure vaccination is also best accomplished with a second-or third-generation vaccine. The second and third generations can be used to protect against the monkeypox virus, although the precise level of protection induced is still under investigation. The two first generations of vaccines are administered by the percutaneous route (scarification), whereas the third-generation vaccines are administered by subcutaneous injection.

The first generation was propagated on the skin of animals (mainly calves, but also sheep and rabbits). Dryvax was a first-generation vaccine (New York City Board of Health strain) produced in the skin of calves, and it was approved in 1931. Rare but serious adverse events (AEs) were observed during large-scale immunization with this vaccine, resulting in one to two deaths per million vaccinees. These adverse effects specifically included self-inoculation, generalized vaccinia, eczema vaccinatum and encephalitis. When Dryvax was used in the early 2000s to vaccinate large numbers of US military personnel and select civilians after a rigorous inclusion screening, fewer serious AEs were observed. Dryvax was revoked as of 29 February 2008. The Elstree vaccine is another first-generation vaccine based on the Lister strain. This vaccine elicited long-term protection [292]. Nevertheless, this vaccine induces a higher rate of serious adverse effects than NYCBH-based vaccines (8.4 vs. 1.4 deaths per million vaccinees).

The second generation consists of plaque-purified live VACV clones propagated in cell culture, resulting in a safer production process. ACAM2000 was a second-generation vaccine, composed of New York City Board of Health strain, approved in 2008. As second-generation vaccines still used live strains, adverse events associated with the vaccine strain were persistent, but they are still used.

To overcome the limits inherent to the use of replicating vaccines, a third generation was developed. Highly attenuated VACV strains were used for the production of this generation of vaccines. Modified vaccinia Ankara (MVA) is a highly attenuated strain, unable to replicate effectively in humans that is used for the development of the Imvanex and Jynneos (Imvamune) vaccines, approved in 2013 and 2019, respectively. In vivo studies demonstrate that MVA is safe, even in NHP and immunodeficient mice models [293,294]. Two doses of MVA, administered 28 days apart, are needed to induce a robust immune response. Non-human primate (NHP) studies have demonstrated that vaccination with MVA induces a comparable immune response and protective efficacy to traditional smallpox vaccines used to eradicate smallpox and protects NHP from severe disease associated with a lethal challenge of the monkeypox virus. Both vaccines are officially approved for the prevention of smallpox and monkeypox infection.

The next generations of smallpox vaccines are under development. The new generation consists of introducing targeted deletions/insertions that disrupt selected viral genes and lead to VACV attenuation via genetic engineering. Additional vaccines are also under development. For example, Sang et al. developed a quadrivalent mRNA vaccine that induces an immune response and protects against the vaccinia virus [295]. This vaccine is composed of mRNA coding for the A29L, A35R, M1R and B6R proteins. BALB/c mice were immunized twice intramuscularly with one of the two quadrivalent mRNA vaccines. Then, mice were challenged with the Tian Tan strain of VACV via the s.c. route. The viral load in the mice was measured at 24 h by bioluminescent assay. The bioluminescent signal was largely undetectable in immunized mice, whereas signal values as high as 6.5 log 10 were detected in naïve mice, which suggests that VACV was rapidly cleared by vaccine-induced antibodies after the challenge. When the sera of vaccinated mice were mixed with the vaccinia virus, and injected into nude mice, a significant neutralization was also observed. Currently, no next-generation vaccines have been approved.

## 9. Concluding Remarks and Perspectives

In the current political and environmental context, the development of medical countermeasures against neglected/rare pathogens, such as potential biowarfare agents, is essential. Antibodies are molecules of choice for the diagnosis and treatment of these diseases. They can overcome the limitations of other molecules such as antibiotics or antivirals, or they can act in synergy with them. Several antibodies have been approved, such as for anthrax therapy. Although the benefits of certain approved antibodies have been disputed, they may, at the very least, contribute to treatment success and at worst, as they are generally very safe, they would not make situations any worse. Antibodies are also of particular interest even when antibiotics are available because they can act in synergy. This represents an alternative treatment against antibiotic resistant strains, as has been observed with Tecovirimat.

One of the limitations of therapeutic antibody use is the potential for eliciting host immune responses. Particularly, anti-drug antibodies (ADA), may prevent repeated or long-term treatments with these molecules. However, unlike the antibodies intended for cancer indications, the treatment of the diseases caused by potential biowarfare agents would require very few antibody injections, in some cases a single dose, thus minimizing the risk of ADA development. Thus, antibodies are particularly suitable in the context of biodefense.

Currently, recombinant antibodies have only been approved for the treatment of anthrax and Ebola, but many antibodies are under development for all targets. Because these diseases are generally rare, very few pharmaceutical companies are developing antibodies. Regarding BoNTs, although the babyBIG immunoglobulin and BAT heptavalent antitoxin are available in the USA and Canada, more efforts are needed for the independent EU production of therapeutic products against BoNT intoxication to protect the European population. Different recombinant antibodies combinations are currently under development, offering an unlimited source of selective agents free of risk of anaphylaxis. To this aim, human antibody phage display libraries have proven their usefulness in providing antibodies-based medical countermeasures. The main challenge is funding the development of pre-clinical and clinical phase 1 development, to traverse the ’valley of death’ between the bench and clinical development. Therapeutic antibodies have better approval success rates during the clinical trials than other molecules. As of November 2023, about 200 antibodies were marketed or are currently in regulatory review in at least one country. In addition, about 1100 therapeutic antibodies were in phase 1, phase 1/2, or phase 2 studies. It is interesting to note that nearly 70% of these are for cancer indications and that antibodies against infectious diseases or toxins are rarer. About 60% of the antibodies under development target novel antigens, suggesting that the biopharmaceutical industry continues to invest in innovative research [30].

## Figures and Tables

**Table 2 microorganisms-12-02622-t002:** Types of VHF, modes of transmission, virulence, therapy and mortality.

Genus	Example of Virus	Transmission to Humans	Natural Human-to-Human Transmission	Indicative ID_50_	Indicative Human R_0_ (if Applicable)	Main Treatment	Mortality (%)
*Bunyaviridae*	Crimean Congo Hemorrhagic Fever	Tick	Yes	ND	<1	Ribavirin (some benefits)	3–40
*Flaviviridae*	Severe dengue	Mosquito	Rare (but mosquitoes can form a cluster): Mother-to-baby: during pregnancy, delivery or breastfeeding	Mosquito ID_50_: 6.29 to 7.52 log 10 RNA copies/mL of plasma	NA	Supportive Prophylaxis: vaccine.	1–20
*Flaviviridae*	Yellow fever	Mosquito	No (but mosquitoes can form a cluster)	8–6.5 log 10 TCID_50_·mL^−1^	NA	Supportive Prophylaxis: vaccine	25–50
*Flaviviridae*	West Nile	Mosquito	Mother-to-baby: during pregnancy, delivery or breastfeeding	Low		Supportive	<1%
*Filoviridae*	Ebola	Primate	Yes	1–10 aerosolized virus particles	1.3–2.53	Supportive Inmazeb (Ebola Zaire) Ansuvimab (Ebola Zaire)	50–90
*Filoviridae*	Marburg	Primate	Yes	1–10 aerosolized virus particles	1.59	Supportive	50–90
*Bunyaviridae*	Hantavirus	Rodent	Yes (but rare)	3 PFU in a hamster model for ANDV	2.12 in a cluster	*Ribavirin (may have some benefits)*	From 1 (Seoul and Pumaala), 5–15 (Hantaan) to 50 (Sin Nombre)
*Bunyaviridae*	Rift Valley Fever	Mosquito	No	Low to moderate		*Ribavirin*	1 RVHF, up to 40 (CCHF)
*Arenaviridae*	Lassa	Rodent	Yes	1–10 aerosolized virus particles	1.23–1.33	Ribavirin	30
*Arenaviridae*	South American Hemorrhagic Fevers	Rodent		Moderate		Supportive	30

Drugs in italic: drugs that are used, at least for some indications, but that are not approved by the FDA or other regulatory agencies. ND: no data. NA: not applicable. It is important to know that virulence and R_0_ may vary significantly depending on the virus strain and the socio-environmental conditions. ID_50_ is also dependent on the route of infection. Adapted from several sources [261,262,263,264,265,266].

## Data Availability

Not applicable.

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
