# Peer review of "Development of Effective Medical Countermeasures Against the Main Biowarfare Agents: The Importance of Antibodies"

_microorganisms, 2024, doi:10.3390/microorganisms12122622_

Round 1

Reviewer 1 Report (Previous Reviewer 2)

Comments and Suggestions for Authors

The authors have reviewed the potential use of therapeutic antibodies to address the current deficiencies in respect to treatments and vaccines for six agents of concern with respect to weaponisation and bioterrorism. The authors have utilised a mix of peer-reviewed publications and some of the grey literature for these pathogens.

Overall the manuscript is well structure and flows well between the descriptions of the pathogens included in the review.

Other points for the authors to consider.

Line 12 Suggest revision “The COVID-19 pandemic has reminded”

The current preferred name for monkeypox is mpox.

The WHO has not declared a pandemic in respect to mpox, rather is has been identified as pathogen of global concern.

Line 6 suggest replacing “6” with “six” as per scientific writing conventions. Similarly, throughout the manuscript where appropriate.

Line 27 consider “therapeutic antibody” as an additional keyword.

Line 57-59 – Please provide an appropriate citation for this statement.

Line 69-72 I would strongly recommend that these statements be deleted. I do not see any value in regurgitating unsubstantiated politically motivated statements in a scientific review of this nature.

Line 75 The title of the table should be above the table. I would recommend that a third column be added with relevant citations for these events where there is one available. Some of the data presented is additional to that in the citated reference.

Lines 78-111 – I see no value in the majority of this text in the context of the topic of this review. I would strongly recommend that the majority of the text be deleted. It is mostly philosophical in nature and not related to the top of this review. It could be condensed to just the gain of function studies.

Line 125 I would suggest replacing “states” with “countries” or “sovereign states”, as in some countries, states have a different definition.

Line 129 Suggest replacing “established” with “recommended”, scientists do not legislate regulations, governments do.

Line 151-155 I would strongly suggest deletion of this text. Lack of evidence is not evidence. With advances in sequencing technology, previously unknown sequences are discovered every day. Add to this the emergence of SARS-CoV-2, could have resulted from spontaneous mutations that increased the likelihood of being able to infect and replicate in humans, there may have been subsequent selection of mutations that resulted in the initial pandemic strain. As we have seen (and has been documented) the virus has continued to evolve in subsequent years, all of which are sequences we have not seen before.

Line 159-162 I would strongly suggest deletion of this text.

Line 172 I would suggest specifying the three countries.

Line 176 see previous comments on mpox. Same comment for other references to the “monkeypox pandemic” for the remainder of the manuscript.

Line 180 I would suggest that “contamination” be replace with “infection” in this context. A person could be contaminated with mpox (eg virus present on the skin or clothing) but not exhibit clinical symptoms, whereas if they are infected with the virus they would be more likely to require clinical intervention.

Line 197 suggest revision “Often, agents associated with neglected / rare diseases are considered potential biowarfare agents, and as result there are few medical countermeasures are available.”

Line 204 207 I would suggest adding some specific details on the better success rate of antibodies in clinical trials. Success compared to what? What are the differences for other technologies compared to 14% to 32% for antibodies?

Line 148 – same comment as previously on the use of “contaminations” in this context.

Line 149 suggest revision “lethality if treated”

Line 630 suggest revision “Clostridium botulinum” as the genus name is not mentioned until line 658. The authors then switch between abbreviating the genus name and not abbreviating it, suggest checking the manuscript carefully to ensure consistency with respect to scientific conventions. Similarly, for other organisms where genus species nomenclature is used.

Line 932 Suggest revision “Its reservoir is small rodents, with humans being accidental hosts that are often infected through flea bites.”

Suggest adding a suitable citation for this statement.

Line 962 Suggest replacing “homology” with “identity”.

Homology is an evolutionary term, two “things” that share a common ancestor, are homologous. It is a yes or no question that cannot be quantified.

Line 964 The acquired genes could be briefly described if their source is know and/or if they are known virulence factors.

Line 1018 The reference does not have an associated citation number.

Line 1024 The reference does not have an associated citation number.

Line 1103 Citation format

Line 1118 The reference does not have an associated citation number.

Lines 1137 to 1146 Please provide an appropriate citation for this text.

Line 1160 Citation format

Line 1416 Citation format

Line 1433 Table 2

Suggest adding an extra column with the citations for these data.

Suggest moving the row with West Nile up so that the flaviruses are sequential.

Line 1621 what does “(21198662=moss 2011)” refer to?

Line 1624 suggest revision “Chemical”

Line 1825: Please add further details that a reader might use to access the same source in the manuscript, e.g., a weblink and the date last accessed.

Line 1843: Please add further details that a reader might use to access the same source in the manuscript, e.g., a weblink and the date last accessed.

Line 1902 Please add further details that a reader might use to access the same source used in the manuscript. Eg weblink and date last accessed.

Line 1903: Please add further details that a reader might use to access the same source in the manuscript, e.g., a weblink and the date last accessed.

Line 2178 Formatting of reference 157.

Line 2255 There seem to be details missing for reference 187.

Line 2361 There seem to be details missing for reference 227.

Author Response

The authors have reviewed the potential use of therapeutic antibodies to address the current deficiencies in respect to treatments and vaccines for six agents of concern with respect to weaponisation and bioterrorism. The authors have utilised a mix of peer-reviewed publications and some of the grey literature for these pathogens.

Overall the manuscript is well structure and flows well between the descriptions of the pathogens included in the review.

Other points for the authors to consider.

Line 12 Suggest revision “The COVID-19 pandemic has reminded”

  • Done

The current preferred name for monkeypox is mpox.

  • Indeed, mpox, now refers to the disease, while monkeypox refers to the virus.

The WHO has not declared a pandemic in respect to mpox, rather is has been identified as pathogen of global concern.

  • Correct: outbreak was used instead of pandemic.

Line 6 suggest replacing “6” with “six” as per scientific writing conventions. Similarly, throughout the manuscript where appropriate.

  • Done for all numbers bellow ten, as recommended by the conventions (i.e. excepted for volumes, quantities…).

Line 27 consider “therapeutic antibody” as an additional keyword.

  • Done

Line 57-59 – Please provide an appropriate citation for this statement.

  • Done

Line 69-72 I would strongly recommend that these statements be deleted. I do not see any value in regurgitating unsubstantiated politically motivated statements in a scientific review of this nature.

  •  

Line 75 The title of the table should be above the table. I would recommend that a third column be added with relevant citations for these events where there is one available. Some of the data presented is additional to that in the citated reference.

  •  

Lines 78-111 – I see no value in the majority of this text in the context of the topic of this review. I would strongly recommend that the majority of the text be deleted. It is mostly philosophical in nature and not related to the top of this review. It could be condensed to just the gain of function studies.

  • Done

Line 125 I would suggest replacing “states” with “countries” or “sovereign states”, as in some countries, states have a different definition.

  • Done

Line 129 Suggest replacing “established” with “recommended”, scientists do not legislate regulations, governments do.

  • Done

Line 151-155 I would strongly suggest deletion of this text. Lack of evidence is not evidence. With advances in sequencing technology, previously unknown sequences are discovered every day. Add to this the emergence of SARS-CoV-2, could have resulted from spontaneous mutations that increased the likelihood of being able to infect and replicate in humans, there may have been subsequent selection of mutations that resulted in the initial pandemic strain. As we have seen (and has been documented) the virus has continued to evolve in subsequent years, all of which are sequences we have not seen before.

  • This is correct. The text was clarified.

Line 159-162 I would strongly suggest deletion of this text.

  • In this section we would like to underline the global risk for public health, caused by the pathogens, whatever the origin is natural or intentional. Nevertheless we modified the text to clarify this point.

Line 172 I would suggest specifying the three countries.

  • Done

Line 176 see previous comments on mpox. Same comment for other references to the “monkeypox pandemic” for the remainder of the manuscript.

  • done

Line 180 I would suggest that “contamination” be replace with “infection” in this context. A person could be contaminated with mpox (eg virus present on the skin or clothing) but not exhibit clinical symptoms, whereas if they are infected with the virus they would be more likely to require clinical intervention.

  • Done

Line 197 suggest revision “Often, agents associated with neglected / rare diseases are considered potential biowarfare agents, and as result there are few medical countermeasures are available.”

  • Not agree. Almost all potential biowarfare agents cause rare diseases, but not all rare diseases are cause by potential biowarfare agents (e.g.: rare cancer…).

Line 204 207 I would suggest adding some specific details on the better success rate of antibodies in clinical trials. Success compared to what? What are the differences for other technologies compared to 14% to 32% for antibodies?

  • We do not used “success” in the text but approval rate. That is to say a molecule that is finally approved vs a molecule that failed to pass de clinical trial.

Line 248 – same comment as previously on the use of “contaminations” in this context.

=> done

Line 249 suggest revision “lethality if treated”

  • Done

Line 630 suggest revision “Clostridium botulinum” as the genus name is not mentioned until line 658. The authors then switch between abbreviating the genus name and not abbreviating it, suggest checking the manuscript carefully to ensure consistency with respect to scientific conventions. Similarly, for other organisms where genus species nomenclature is used.

  • Done

Line 932 Suggest revision “Its reservoir is small rodents, with humans being accidental hosts that are often infected through flea bites.”

  • Done

Suggest adding a suitable citation for this statement.

  • Done

Line 962 Suggest replacing “homology” with “identity”.

  • Done

Homology is an evolutionary term, two “things” that share a common ancestor, are homologous. It is a yes or no question that cannot be quantified.

Line 964 The acquired genes could be briefly described if their source is know and/or if they are known virulence factors.

  •  

Line 1018 The reference does not have an associated citation number.

=>Done.

Line 1024 The reference does not have an associated citation number.

=>Done.

Line 1103 Citation format

=>Done.

Line 1118 The reference does not have an associated citation number.

=>Done.

Lines 1137 to 1146 Please provide an appropriate citation for this text.

=>Done

Line 1160 Citation format

=>Done

Line 1416 Citation format

=>Done

Line 1433 Table 2

Suggest adding an extra column with the citations for these data.

=>The legend was modified. No column was added because all lines merged the same references, so we have keep the reference in the legend, but clarified the legend.

Suggest moving the row with West Nile up so that the flaviruses are sequential.

=>Done

Line 1621 what does “(21198662=moss 2011)” refer to?

  • Done ; reference modified.

Line 1624 suggest revision “Chemical”

  • Done

Line 1825: Please add further details that a reader might use to access the same source in the manuscript, e.g., a weblink and the date last accessed.

=>Done

Line 1843: Please add further details that a reader might use to access the same source in the manuscript, e.g., a weblink and the date last accessed.

  • Done

Line 1902 Please add further details that a reader might use to access the same source used in the manuscript. Eg weblink and date last accessed.

  • Done

Line 1903: Please add further details that a reader might use to access the same source in the manuscript, e.g., a weblink and the date last accessed.

  • Done

Line 2178 Formatting of reference 157.

=>Done.

Line 2255 There seem to be details missing for reference 187.

=>Done

Line 2361 There seem to be details missing for reference 227.

=>Done.

Reviewer 2 Report (New Reviewer)

Comments and Suggestions for Authors

Seven "biowarfare agents" are described in the review, including the properties of individual biowarfare agent and their medical countermeasures.  The following issues should be addressed before further processing.

1. The definition of "Biowarfare agents" should be provided in the INTRODUCTION.

2. The events described in the context necessitate supporting references as empirical evidence, such as: Russia has accused Ukraine and the United States of developing biological weapons and called for an international investigation (Lines 70-71), and also those in Lines 213-218.

3. Syntax errors should be corrected carefully, such as “. Cutaneous anthrax represents is at the origin of almost all natural contaminations” (line 248).

Author Response

Seven "biowarfare agents" are described in the review, including the properties of individual biowarfare agent and their medical countermeasures.  The following issues should be addressed before further processing.

  1. The definition of "Biowarfare agents" should be provided in the INTRODUCTION.

=> Done

  1. The events described in the context necessitate supporting references as empirical evidence, such as: Russia has accused Ukraine and the United States of developing biological weapons and called for an international investigation (Lines 70-71), and also those in Lines 213-218.

=> Done

  1. Syntax errors should be corrected carefully, such as “. Cutaneous anthrax represents is at the origin of almost all natural contaminations” (line 248).

=>Done

Round 2

Reviewer 1 Report (Previous Reviewer 2)

Comments and Suggestions for Authors

The authors have addressed most of the comments and suggestions I made on the submitted version of their manuscript.

 Line 12 This refers to the current mpox outbreaks as a pandemic. As mentioned in my previous review (and amended elsewhere in the manuscript), mpox is considered a pathogen of concern by the WHO, not declared as pandemic. Suggest revision.

Lines 147-162 I acknowledge that the authors have revised this text considerably. However, still think it requires further improvement and balance. The paragraph starts of with a statement about the origins of the COVID-19 pandemic, and five references are cited to support it, Refs 19-24. All of the cited references are from 2021, with one from 2022, all early in the pandemic from an investigative point of view. 

From a balanced perspective, there have been studies since this time that have addressed these issues in detail. I would suggest the authors review some of the more recent literature and consider providing

For example, https://journals.asm.org/doi/10.1128/mbio.00583-23 - provides an excellent discussion of the various hypotheses around the emergence of SARS-CoV-2 from multiple perspectives. 

I am not associated with this editorial nor an author of any of the manuscripts it cites.

Author Response

The authors have addressed most of the comments and suggestions I made on the submitted version of their manuscript.

 Line 12 This refers to the current mpox outbreaks as a pandemic. As mentioned in my previous review (and amended elsewhere in the manuscript), mpox is considered a pathogen of concern by the WHO, not declared as pandemic. Suggest revision.

  • Done. Now, we used the word “crisis” in this sentence.

Lines 147-162 I acknowledge that the authors have revised this text considerably. However, still think it requires further improvement and balance. The paragraph starts of with a statement about the origins of the COVID-19 pandemic, and five references are cited to support it, Refs 19-24. All of the cited references are from 2021, with one from 2022, all early in the pandemic from an investigative point of view. 

From a balanced perspective, there have been studies since this time that have addressed these issues in detail. I would suggest the authors review some of the more recent literature and consider providing

For example, https://journals.asm.org/doi/10.1128/mbio.00583-23 - provides an excellent discussion of the various hypotheses around the emergence of SARS-CoV-2 from multiple perspectives. 

I am not associated with this editorial nor an author of any of the manuscripts it cites.

  • Done. The suggested reference was added, and other, more recent citations, were also added. References added are in favor of both virus-origin. We modified the text: we removed a lot of details concerning the origin of SARS-CoV-2. Indeed, the main idea of this paragraph is to underline the consequences of a biological agent, whatever it was of natural, accidental or intentional origin. A lot of international committees published data concerning the origin of SARS-CoV-2, and we cannot summarize them rigorously in only one paragraph. We just kept a generic paragraph that explain that several hypotheses are still under investigation.

This manuscript is a resubmission of an earlier submission. The following is a list of the peer review reports and author responses from that submission.

Round 1

Reviewer 1 Report

Comments and Suggestions for Authors

The review article by Avri, Gullier, and Escargueil regarding medical countermeasures for biowarfare agents is extremely ambitious.  Because the enormous subject matter, the article is overwhelming, inconsistent in its organization, and reads more like a dissertation or textbook section. While in general it is well written, I believe in its current state it would be difficult for many readers to find the information useful. The following concerns should be addressed prior to publication:

MAJOR Issues:

1.       This review article is an inconsistent combination of opinion and interpretation. The article should be consolidated and more focused on a smaller subject area. Consider using more and consistent subheadings to keep the information on multiple biological agents consistent

2.       While this review article does cite many references, I question the relevance and placement of these citations in many cases. The references appear to be mainly recent, and a great deal of early seminal work is not described. References should be distributed throughout the writing and not at the end of large paragraphs.

MINOR Issues (These are just a few because of the length of the paper, but consistency is lacking throughout the manuscript, for example vaccine are discussed for some agents, PPE for some agents, etc.):

1.Line 30 it is redundant to say Biological agent and Bioweapon, I would change to “Biological agents as weapons”

2. Line 40, again redundant to say, “nations initiated national programs”, change to “nations initiated programs”

3. Lines 42-60. This section comes across as a narrative largely based in “hearsay”, appropriate references should be placed throughout the paragraph (and manuscript in general) at appropriate points to indicate what data the conclusions and summaries are based upon.

4. Line 62: In the US anthrax attack, the PRESS was also a main target and not just governmental officials, lending further to the “bioterror” aspect.

5. Lines 65-68: The authors may be over interpreting the FBI report. To say access to select agents is “easy” (even in the early 2000s) is inaccurate and debatable. I would also suggest reconsidering naming the alleged “prime suspect”, again this information remains debated, is the subject of multiple documentaries and adds little if anything to this review.

6. Line 69-70 this sentence should be removed, the fact that this review is choosing to cite Russian disinformation makes me uncomfortable and, in my opinion, detracts from the overall work.

7. Table 1. The authors should be consistent with using titles, if an individual was a US president they should always be referred to in that manner, this goes for all titles throughout the manuscript.

8. Line 91, this sentence is confusing, and I am unclear to the intent “ Research into…effective treatments”

9. Line 188-201, this paragraph comes across and opinion/commentary in some regard, It should be reexamined and reinforced with appropriate references.  Who indicates that “Antibodies should be the molecules of choice”? Cite more data indicating that antibodies have a  greater success rate.

10. Line 217, as discussed alter some polyclonals are human derived (anthrax) but in tis section it indicates that animal derivation is the largest weakness.

11. Line 219. Is “cocktail” clearer that “oligoclonal”

12. Line 229 (and throughout) the current naming convention is “gram-negative” and not “Gram-negative”…”Gram should only be used referencing the technique or actual stain.

13. Line 245 (and throughout). Genetic nomenclature for genes and plasmids should be consistent.  This should be pXO1 and gene names should be italicized throughout.

14. Line 252-266, this is a summary of decades of work with no references, whenever possible original research should be cited for these important scientific discoveries

15.  Line 294, this is a repeat concept from earlier, please make concise and non-repetitive

16. Line 298, anthrax is not aerosolized. Anthrax is the disease caused by inhalation of Bacillus anthracis spores.

17. Lines 352-372 this section needs more references for example new CDC guidelines may not always require 60 days of antibiotics, cite refs for treatment strategies. In some cases, contaminated should be replaced with infected. And gene names need to be italicized.

18. there are some papers describing antibiotic resistant Bacillus anthracis outside of the lab.

Zhang W, Suyamud B, Lohwacharin J, Yang Y. Large-scale pattern of resistance genes and bacterial community in the tap water along the middle and low reaches of the Yangtze River. Ecotoxicol Environ Saf. 2021 Jan 15;208:111517.

Muriuki SW, Neondo JO, Budambula NLM. Detection and Profiling of Antibiotic Resistance among Culturable Bacterial Isolates in Vended Food and Soil Samples. Int J Microbiol. 2020 Sep 4;2020:6572693.

  Bakici MZ, Elaldi N, Bakir M, DökmetaÅŸ I, Erandaç M, Turan M. Antimicrobial susceptibility of Bacillus anthracis in an endemic area. Scand J Infect Dis. 2002;34(8):564-6.

19. Line 408 and throughout, make sure abbreviations are all spelled out initially such as IM.

20. Line 414 and throughout, it is not necessary repeat Bacillus anthracis after the first spelling but rather B. anthracis is appropriate.

21. Line 447 “only” is distracting, be straightforward and descriptive in word choice.

22. Line 465, what are some examples of ricin derived products

23. Line 503. Define “hot zones”

24. Line 521 define I.p. and be consistent with route descriptions, i.e, IM, IP, etc.

25. Line 537, letter referring to stats should be in italics (p)

26. Line 543, now you are writing out the routes completely

27. Line 548. Accurately define VHH

28. Line 578, the description of ricin vaccines only being useful to select groups could be applied to many of these agents.

29. Line 583 spell out USAMRIID

30. Line 625 already spelled out CDC no need to do again. Also please revaluate the use of Category A and B agents…the CDC now uses “Tiers” to describe select agents.  Correct throughout.

31. Lines 650-655 should be reevaluated, and the use of the term “Nowadays” is irrelevant, as this would be a terror incident regardless of the timeframe.

32. Line 668.  Would “assessed” or “evaluated” be more appropriate than “approached”

33. Line 967, should be “low-calcium response”

34. Line 974, why “pseudocapsule”?

35. Line 978 would gold standard: be better than “reference examination”?

35. Lines 1004-1006 why are the authors talking about PPE for plague not the others, should be consistent for this article to be useful and focused.

36. Lines 1106-1109  an anti-LcrV response is quite protective across multiple platforms…please explain the “less promising” concept

37. Line 1118, was alhydrogel replaced or was CpG added to alhydrogel?

38. Line 1157 there are other more recent reference demonstrating the utility of live attenuated strains to protect against pneumonic plague.

39. Line 1167. Based upon significant data, tularemia is not one of the deadliest, while I agree that it has a low infectious dose, its mortality rates do not warrant the “most deadly” designation.

40. Line 1268. The authors should address the regulatory requirements for diagnostic samples compared to research samples.

41. Line 1588  should be BSL4

Comments on the Quality of English Language

English is mostly fine, editing required

Author Response

The review article by Avri, Gullier, and Escargueil regarding medical countermeasures for biowarfare agents is extremely ambitious.  Because the enormous subject matter, the article is overwhelming, inconsistent in its organization, and reads more like a dissertation or textbook section. While in general it is well written, I believe in its current state it would be difficult for many readers to find the information useful. The following concerns should be addressed prior to publication:

=>We are not agree with the reviewer. This review is ambitious, but it will not replace a dedicated review for each pathogen. Nevertheless, in a single review we successfully provided all up to date information concerning the medical countermeasures against biowarfare agent. The objective of this review is to alert the scientific community concerning the urgent need for the development of new medical countermeasure for these neglected pathogens. The review is long, but a specific chapter is dedicated for each pathogen. In each chapter, a similar organization is used (presentation of each class of therapeutic molecules).

MAJOR Issues:

  1. This review article is an inconsistent combination of opinion and interpretation. The article should be consolidated and more focused on a smaller subject area. Consider using more and consistent subheadings to keep the information on multiple biological agents consistent

=>We are not agree the originality and the strength of this review is the presentation of all major biowarfare agent and not only one. Several review were already available for each of the pathogen, but almost no review present the biowarfare agent in their globality. By compiling the information in a single review, it is more easy to underline the urgent need for the development of new medical countermeasure. Is also make possible the identification of similar needs for several pathogens.

  1. While this review article does cite many references, I question the relevance and placement of these citations in many cases. The references appear to be mainly recent, and a great deal of early seminal work is not described. References should be distributed throughout the writing and not at the end of large paragraphs.

=>In this review we added both historical and recent references. They are correctly placed, and not only at the end of the paragraphs. Only the most relevant references were added. It is true that not all works were not presented, because a lot of them is not sufficiently advanced. Such “preliminary data” could only be the subject of an extensive review for the specific pathogen.

MINOR Issues (These are just a few because of the length of the paper, but consistency is lacking throughout the manuscript, for example vaccine are discussed for some agents, PPE for some agents, etc.):

=>It is normal because sometime no vaccines were approved of sufficiently advanced to be presented.

1.Line 30 it is redundant to say Biological agent and Bioweapon, I would change to “Biological agents as weapons”

=>It is already the case.

  1. Line 40, again redundant to say, “nations initiated national programs”, change to “nations initiated programs”

=>Not agree national is used by opposition to international or collaborative. In this context that mean that a nation developed an “internal” program.

  1. Lines 42-60. This section comes across as a narrative largely based in “hearsay”, appropriate references should be placed throughout the paragraph (and manuscript in general) at appropriate points to indicate what data the conclusions and summaries are based upon.

=>They are not “hearsay”. The facts presented are based on national and international organization inspection and reports. These reports are official, even in some case they concluded that de weaponization was not observed, but that all the preliminary steps for weaponization were realized. In this section, the references are not at the end. For example, a reference is placed after le first sentence. Refences are on lines 42, 50, 55, and 57, that is to say after each two sentences in average in the section 42-60.

  1. Line 62: In the US anthrax attack, the PRESS was also a main target and not just governmental officials, lending further to the “bioterror” aspect.

=>Correct. Precision added.

  1. Lines 65-68: The authors may be over interpreting the FBI report. To say access to select agents is “easy” (even in the early 2000s) is inaccurate and debatable. I would also suggest reconsidering naming the alleged “prime suspect”, again this information remains debated, is the subject of multiple documentaries and adds little if anything to this review.

=>The implication of Bruce Ivins is still discussed by some people. Nevertheless, in this review we only referred to official facts. The FBI definitively considered Bruce Ivins as guilty and actually not only as a prime suspect. Despite this, we still used prime suspect, because no court has sentenced Bruce IVINS.  Nevertheless we modified slightly the sentence (e.g. : relatively easy…).

  1. Line 69-70 this sentence should be removed, the fact that this review is choosing to cite Russian disinformation makes me uncomfortable and, in my opinion, detracts from the overall work.

=>It is not up to us to determine if the accusation could be assimilated or not to disinformation. We only say that Russia officially requested an inspection of UN. Nevertheless, we add a “disclaimer”. 

  1. Table 1. The authors should be consistent with using titles, if an individual was a US president they should always be referred to in that manner, this goes for all titles throughout the manuscript.

=>Done

  1. Line 91, this sentence is confusing, and I am unclear to the intent “ Research into…effective treatments”

=>Sentence modified.

  1. Line 188-201, this paragraph comes across and opinion/commentary in some regard, It should be reexamined and reinforced with appropriate references.  Who indicates that “Antibodies should be the molecules of choice”? Cite more data indicating that antibodies have a  greater success rate.

=>Not agree. A reference showing that antibodies have an higher success rate was quote in the text. We consider that only one reference is sufficient, is this reference is relevant and representative of the other ones. Nevertheless, three new references were added.

  1. Line 217, as discussed alter some polyclonals are human derived (anthrax) but in tis section it indicates that animal derivation is the largest weakness.

=>Polyclonal from humans are rare. But indeed, we modified the sentence: “generally are of animal origin”. We also add a sentence concerning the limitation of human polyclonal antibodies.

  1. Line 219. Is “cocktail” clearer that “oligoclonal”

=>Both word are generally used

  1. Line 229 (and throughout) the current naming convention is “gram-negative” and not “Gram-negative”…”Gram should only be used referencing the technique or actual stain.

=>To our knowledge Gram-negative is the correct way.

  1. Line 245 (and throughout). Genetic nomenclature for genes and plasmids should be consistent.  This should be pXO1 and gene names should be italicized throughout.

=>Correct, done.

  1. Line 252-266, this is a summary of decades of work with no references, whenever possible original research should be cited for these important scientific discoveries

=>These works are not the topic of this review. The date presented in 252-266 section are described in the publication quoted before.

  1. Line 294, this is a repeat concept from earlier, please make concise and non-repetitive

=>Done

  1. Line 298, anthrax is not aerosolized. Anthrax is the disease caused by inhalation of Bacillus anthracis spores.

=>The text is correct, because this is not anthrax, but anthrax ames spores that are aerosolized.

  1. Lines 352-372 this section needs more references for example new CDC guidelines may not always require 60 days of antibiotics, cite refs for treatment strategies. In some cases, contaminated should be replaced with infected. And gene names need to be italicized.

=>Still 60 days. Ok for gene in italic.

  1. there are some papers describing antibiotic resistant Bacillus anthracis outside of the lab.

Zhang W, Suyamud B, Lohwacharin J, Yang Y. Large-scale pattern of resistance genes and bacterial community in the tap water along the middle and low reaches of the Yangtze River. Ecotoxicol Environ Saf. 2021 Jan 15;208:111517.

Muriuki SW, Neondo JO, Budambula NLM. Detection and Profiling of Antibiotic Resistance among Culturable Bacterial Isolates in Vended Food and Soil Samples. Int J Microbiol. 2020 Sep 4;2020:6572693.

  Bakici MZ, Elaldi N, Bakir M, DökmetaÅŸ I, Erandaç M, Turan M. Antimicrobial susceptibility of Bacillus anthracis in an endemic area. Scand J Infect Dis. 2002;34(8):564-6.

=>Yes, we said that it is rare and not inexistant. A reference is already in the text and we consider that it is enough. This is not a review completely dedicated to anthrax.

  1. Line 408 and throughout, make sure abbreviations are all spelled out initially such as IM.

=>Done

  1. Line 414 and throughout, it is not necessary repeat Bacillus anthracis after the first spelling but rather B. anthracis is appropriate.

=>Done.

  1. Line 447 “only” is distracting, be straightforward and descriptive in word choice.

=>Done.

  1. Line 465, what are some examples of ricin derived products

=>This is already in the text, bellow.

  1. Line 503. Define “hot zones”

=>We think that it is not necessary.

  1. Line 521 define I.p. and be consistent with route descriptions, i.e, IM, IP, etc.

=>Done

  1. Line 537, letter referring to stats should be in italics (p)

=>Done

  1. Line 543, now you are writing out the routes completely

=>?

  1. Line 548. Accurately define VHH

=>This is defined.

  1. Line 578, the description of ricin vaccines only being useful to select groups could be applied to many of these agents.

=>Yes.

  1. Line 583 spell out USAMRIID

=>Done

  1. Line 625 already spelled out CDC no need to do again. Also please revaluate the use of Category A and B agents…the CDC now uses “Tiers” to describe select agents.  Correct throughout.

=>Done.

  1. Lines 650-655 should be reevaluated, and the use of the term “Nowadays” is irrelevant, as this would be a terror incident regardless of the timeframe.

=>Done

  1. Line 668.  Would “assessed” or “evaluated” be more appropriate than “approached”

=>Done

  1. Line 967, should be “low-calcium response”

=>Done

  1. Line 974, why “pseudocapsule”?

=>Done

  1. Line 978 would gold standard: be better than “reference examination”?

=>Done

  1. Lines 1004-1006 why are the authors talking about PPE for plague not the others, should be consistent for this article to be useful and focused.

=>Removed.

  1. Lines 1106-1109  an anti-LcrV response is quite protective across multiple platforms…please explain the “less promising” concept

=>done

  1. Line 1118, was alhydrogel replaced or was CpG added to alhydrogel?

=>done

  1. Line 1157 there are other more recent reference demonstrating the utility of live attenuated strains to protect against pneumonic plague.

=>done

  1. Line 1167. Based upon significant data, tularemia is not one of the deadliest, while I agree that it has a low infectious dose, its mortality rates do not warrant the “most deadly” designation.

=>agree.

  1. Line 1268. The authors should address the regulatory requirements for diagnostic samples compared to research samples.

=>done

  1. Line 1588  should be BSL4

=>correct.

Reviewer 2 Report

Comments and Suggestions for Authors

The authors have reviewed various agents with the potential to be used as biological warfare agents. After a long introduction that describes the history of these agents through human history using both peer-review and grey literature sources, the authors state the review aims are to give an overview of current therapeutics for each to demonstrate that the development of additional therapeutic antibodies is essential for public health. I am not convinced that the authors achieve this aim. The final conclusions section is relatively brief, certainly compared to the remainder of the manuscript, but does not answer the state aims. Similarly, it ends with a potential drawback of the use of therapeutic antibodies. I think it would be better to end on a positive note regarding the use of antibodies as therapeutic agents. The last few sentences seem like an after thought.

Overall, I felt the main text was a little disjointed and would suggest the authors review the structure of their manuscript. This would include standardising the headings and their order between the different sections on each pathogen. This would enable a reader with specific interests to seamlessly find the sections they are interested in. Also a more streamlined structure would improve the readability of a manuscript of this length.

I would also suggest that the authors review each section to ensure they have provided their perspective on the studies they have decided to include, weakness, strengths, and informed future research directions. Some might suggest that they have included too much detail, however I support the inclusion of specific details in reviews. Otherwise, the review can end up as a bunch of statements with no details. However, it should also be clear why a particular study was included in the context of the review subject, particularly why it is important to the field. The presented details also need to support the position posed by the authors.

Comments and suggestions:

Line 15 suggest replacing “destabilizing the economy” with “negatively impacting on the economy”, as destabilize is used previously in the sentence.

Line 43 suggest revision “estimated at US$1 billion”, or similar as long as the currency is specified. See line 213.

Line 20 suggest revision “antibodies and chemical drugs”

Lines 67-68 Are the authors able to provide a suitable reference for this statement?

Line 85-87 I would argue that the role of ethics committees is not to avoid publication of controversial data, rather their role is to ensure that research is conducted in an ethical manner (scientific and community expectations) within the laws of the jurisdiction where it takes place.

Lines 87-93 I would suggest that the authors include primary citations for these gain of function studies, rather than citing papers that critique/comment on these papers.

Lines 103-105 I think these comments require additional evidence. While technically it might seem to be feasible to use artificial intelligence in this way, I am not convinced that it makes the process easier. Moving from a concept to practical implementation, particularly with the pathogens of interest, to the point where it is a potential biological weapon is not a trivial process.

Line 114 please specific the number of countries.

Line 126 I would suggest review of this statement to ensure clarity. Not all GMOs would be considered dangerous biological agents (eg a transgenic zebrafish). Does the statement refer to genetically modified pathogens?

Perhaps modification to something along the lines of:

“confinement of dangerous biological agents, such as those that have undergone genetic modification, were given.”

However, even this is not overly accurate to me, as genetic modification could be used to attenuate these pathogens as well.

Line 146 to 157

In addition, the authors would most likely have heard the idiom aphorism "Absence of evidence is not evidence of absence" or variations thereof. Thus that there are inclusive reports of zoonotic transfer, does not mean that it was not the source. What is more likely, in my opinion, to have led to these inconclusive reports is a lack of cooperation and transparency from the country where (most likely) SARS-CoV2 originated from

The text is also tautology, the authors first state “strong evidence to support zoonotic transfer” then “inconclusive reports” contradicting this. So I would ask, on the weight of evidence what would the authors conclude?

Line 175 suggest replacing “contaminations” with “infections”

Line 196 I would suggest adding a relevant citation(s) regarding the first approved therapeutic antibody.

Line 222 I would suggest splitting the aim of the review into its own section, 1.6.

Line 223 I would rewording this sentence to delete “brief comparison”. There is nothing “brief” about the manuscript and I do not think the authors directly “compare” the antibody-based treatments of the agents included in the manuscript.

Line 230 suggest replacing “contaminate” with “infect”

Line 240 suggest deletion of “(BBB)” it does not appear to be used again the manuscript.

Line 262 suggest revision “drug-addiction”

Line 283 suggest revision “Zealand”

Line 295 The term “New Zealand White Rabbit” was abbreviated on line 283

Line 369-371 This sentence seems to contradict the previous text which suggests that poor compliance with taking the antimicrobials resulted in the infections and deaths, not that the drugs were ineffective. Please review.

Line 476 suggest revision “and 2018, respectively.”

Line 476 and 478 I would suggest deleting the nationalities of these individuals. Their affiliations with Daesh is more pertinent in this context.

Line 507 suggest revision “Yu et al. [67] isolated”

Line 575 I think it is erroneous to refer to ricin as a “pathogen”, similarly it is not a disease.

Consider “As a noninfectious agent associated with toxicity”

Line 616 Please provide a suitable citation(s) to support this statement.

Line 647 suggest revision “A volume of 19,00 L”

Line 647-648 Please provide a suitable citation(s) to support this statement on the toxic potential of this quantity of BoNT.

Line 1438 Table 2 – The table format is different from Table 1.

The abbreviated example viruses should be provided in full, either in the title or as footnotes.

Suggest moving the West Nile Virus row up the table so that all the flaviviruses are together.

Line 1756-1757 Please review the sentence:

“This represents an alternative treatment against antibiotic-resistant strains, as has been observed with Tecovirimat.”

Based on the previous text, tecovirimat is a potential treatment of smallpox, for which antibiotic treatment would be irrelevant.

Line 1770 suggest revision “One of the limitations of therapeutic antibody use is the potential for eliciting host immune responses, anti-drug antibodies (ADA), that may prevent repeated or long-term treatments with these molecules.

Line 1774 suggest revision “in some cases a single dose, thus minimizing the risk of ADA development.

Comments on the Quality of English Language

See comments to authors. Nothing major some misuse of terms here and there.

Author Response

The authors have reviewed various agents with the potential to be used as biological warfare agents. After a long introduction that describes the history of these agents through human history using both peer-review and grey literature sources, the authors state the review aims are to give an overview of current therapeutics for each to demonstrate that the development of additional therapeutic antibodies is essential for public health. I am not convinced that the authors achieve this aim. The final conclusions section is relatively brief, certainly compared to the remainder of the manuscript, but does not answer the state aims. Similarly, it ends with a potential drawback of the use of therapeutic antibodies. I think it would be better to end on a positive note regarding the use of antibodies as therapeutic agents. The last few sentences seem like an after thought.

=>We are not agreed. We think that we reach our aim. Indeed, in a single review we provided an extensive overview of each major pathogen. Each pathogen could be the subject of a dedicated review. Here, we have successfully provided sufficient data to present extensively that state of the art.

=>Antibody are molecules of particular interest, but they have drawbacks that are described. But actually we modified the final section to finish is a most positive way.

Overall, I felt the main text was a little disjointed and would suggest the authors review the structure of their manuscript. This would include standardising the headings and their order between the different sections on each pathogen. This would enable a reader with specific interests to seamlessly find the sections they are interested in. Also a more streamlined structure would improve the readability of a manuscript of this length.

=>Done.

I would also suggest that the authors review each section to ensure they have provided their perspective on the studies they have decided to include, weakness, strengths, and informed future research directions. Some might suggest that they have included too much detail, however I support the inclusion of specific details in reviews. Otherwise, the review can end up as a bunch of statements with no details. However, it should also be clear why a particular study was included in the context of the review subject, particularly why it is important to the field. The presented details also need to support the position posed by the authors.

=>In the introduction and in the conclusion we explained that we focus on antibodies because we think they are molecules of choice for biodefense. We also provided data concerning other molecules (antiviral, antibiotics…) to show the weakness of this molecules or the potential synergy with antibodies. Introduction and conclusion was modified to clarified it.

Line 15 suggest replacing “destabilizing the economy” with “negatively impacting on the economy”, as destabilize is used previously in the sentence.

=>Done

Line 43 suggest revision “estimated at US$1 billion”, or similar as long as the currency is specified. See line 213.

=>Done

Line 20 suggest revision “antibodies and chemical drugs”

=>Done

Lines 67-68 Are the authors able to provide a suitable reference for this statement?

=>Done

Line 85-87 I would argue that the role of ethics committees is not to avoid publication of controversial data, rather their role is to ensure that research is conducted in an ethical manner (scientific and community expectations) within the laws of the jurisdiction where it takes place.

=>Not completely agree. Yes, the role of ethical committees of to ensure that research is conducted in an ethical manner. But, dual side research programs can be considers are not ethical and some time should not be published. In addition NSABB is not a conventional ethical committee ; it is a committee that evaluate the consequences for the biosecurity. Sentence not modified in the manuscript.

Lines 87-93 I would suggest that the authors include primary citations for these gain of function studies, rather than citing papers that critique/comment on these papers.

  • It was already the case, but an additional reference was added.

Lines 103-105 I think these comments require additional evidence. While technically it might seem to be feasible to use artificial intelligence in this way, I am not convinced that it makes the process easier. Moving from a concept to practical implementation, particularly with the pathogens of interest, to the point where it is a potential biological weapon is not a trivial process.

  • The risk of AI for the biosecurity is not the topic of this review. It could be the subject of a full review. The data available are limited but several scientist and government consider AI as a factor that facilitate the work of terrorist. AI is already used, with success, by several pharmaceutical companied, wich prove its efficacy.

Line 114 please specific the number of countries.

  • The number of countries that signed in 1972 is not relevant. Indeed, each country have to sign the ratify the convention. This process could be long. In 1975, 22 countries signed and ratified the convention. The number of countries that signed and/or ratified the text, today, is already specified in the text.

Line 126 I would suggest review of this statement to ensure clarity. Not all GMOs would be considered dangerous biological agents (eg a transgenic zebrafish). Does the statement refer to genetically modified pathogens?

Perhaps modification to something along the lines of:

“confinement of dangerous biological agents, such as those that have undergone genetic modification, were given.”

However, even this is not overly accurate to me, as genetic modification could be used to attenuate these pathogens as well.

=>All GMO are potentially dangerous. Indeed, by definition, they are not natural. The notion of danger is not only limited to bioterrorism, but also for the environment, for example. For all GMO, the level of danger have to be considered before its development. It is also important to not that in the 70s, the level of knowledge concerning the GMO were not the same that today, so they were initially considered as dangerous. A sentence was added in the text to clarify it.

Line 146 to 157

In addition, the authors would most likely have heard the idiom aphorism "Absence of evidence is not evidence of absence" or variations thereof. Thus that there are inclusive reports of zoonotic transfer, does not mean that it was not the source. What is more likely, in my opinion, to have led to these inconclusive reports is a lack of cooperation and transparency from the country where (most likely) SARS-CoV2 originated from

The text is also tautology, the authors first state “strong evidence to support zoonotic transfer” then “inconclusive reports” contradicting this. So I would ask, on the weight of evidence what would the authors conclude?

=>The topic of this review is not to make conclusion about SARS-CoV-2 genesis. A lot of scientists and governmental institutions are working on it and all these experts still not have formal conclusions. The objective of this paragraph is to underlined of biological agent can destabilize the world, whatever the pathogen appears naturally or not.

Line 175 suggest replacing “contaminations” with “infections”

=>Not agree. Contamination refers to the penetration of a pathogen in an organism. Infection refers to the multiplication of the pathogen into the organism. In the sentence, contamination is the correct word.

Line 196 I would suggest adding a relevant citation(s) regarding the first approved therapeutic antibody.

=>Done

Line 222 I would suggest splitting the aim of the review into its own section, 1.6.

=>Done.

Line 223 I would rewording this sentence to delete “brief comparison”. There is nothing “brief” about the manuscript and I do not think the authors directly “compare” the antibody-based treatments of the agents included in the manuscript.

=>Done.

Line 230 suggest replacing “contaminate” with “infect”

=>Done

Line 240 suggest deletion of “(BBB)” it does not appear to be used again the manuscript.

=>Done

Line 262 suggest revision “drug-addiction”

=>Not agree, according to NIH, drug-abuse is the correct word. Drug-addiction refers to the addiction and not only to the consummation of drugs.

Line 283 suggest revision “Zealand”

=>Done.

Line 295 The term “New Zealand White Rabbit” was abbreviated on line 283

=>Done.

Line 369-371 This sentence seems to contradict the previous text which suggests that poor compliance with taking the antimicrobials resulted in the infections and deaths, not that the drugs were ineffective. Please review.

=>Not agree. This is not a contradiction. Both constats are independent. The first one is that compliance is low, because antibiotic treatment is very long. The second one is that, even antibiotic treatment is correctly administrated, this is not a guarantee of survival. Indeed, in some cases antibiotic treatment is started to late or because the level of pathogen is too high.

Line 476 suggest revision “and 2018, respectively.”

=>Done.

Line 476 and 478 I would suggest deleting the nationalities of these individuals. Their affiliations with Daesh is more pertinent in this context.

=>I agree. Done.

Line 507 suggest revision “Yu et al. [67] isolated”

=>Done.

Line 575 I think it is erroneous to refer to ricin as a “pathogen”, similarly it is not a disease.

Consider “As a noninfectious agent associated with toxicity”

=>Not agree. Pathogen refers to patho-genesis, so to a substance that can induce a disease.

Line 616 Please provide a suitable citation(s) to support this statement.

=>Done

Line 647 suggest revision “A volume of 19,00 L”

=>No. 19,000 refers to 19000L and not to 19L.

Line 647-648 Please provide a suitable citation(s) to support this statement on the toxic potential of this quantity of BoNT.

=>The calculation is easy and will not be added in the text (the volume is 19000L and not 19L). This volume can kill 21,111,111,111 humans.

Line 1438 Table 2 – The table format is different from Table 1.

The abbreviated example viruses should be provided in full, either in the title or as footnotes.

Suggest moving the West Nile Virus row up the table so that all the flaviviruses are together.

Line 1756-1757 Please review the sentence:

“This represents an alternative treatment against antibiotic-resistant strains, as has been observed with Tecovirimat.”

=>Done.

Based on the previous text, tecovirimat is a potential treatment of smallpox, for which antibiotic treatment would be irrelevant.

Line 1770 suggest revision “One of the limitations of therapeutic antibody use is the potential for eliciting host immune responses, anti-drug antibodies (ADA), that may prevent repeated or long-term treatments with these molecules.

=>Done.

Line 1774 suggest revision “in some cases a single dose, thus minimizing the risk of ADA development.

=>Done.

Round 2

Reviewer 1 Report

Comments and Suggestions for Authors

This review is too large to be meaningful and borders on opinion in some places.  The authors did not take the review process seriously and I am recommending rejection of this manuscript.

Comments on the Quality of English Language

minor edits required

Reviewer 2 Report

Comments and Suggestions for Authors

The authors have addressed the comments and suggestions I made in the original review of their manuscript. 

I have no comments on the revised version of this review.